# Assessing the large-scale plant-water relations in the humid subtropical Pearl River Basin of China

Hailong Wang[1, 2, 3], Kai Duan[1, 2, 3], Bingjun Liu[1, 2], Xiaohong Chen[1, 2]

[1] School of Civil Engineering, Sun Yat-sen University, Guangzhou, Guangdong, China, 510275

[2] Guangdong Engineering Technology Research Center of Water Security Regulation and Control for Southern China, Sun Yat-sen University, Guangzhou, China, 510275

[3] Southern Marine Science and Engineering Guangdong Laboratory, Zhuhai, Guangdong, China, 519082

*Correspondence to*: Hailong Wang (wanghlong3@mail.sysu.edu.cn; whl84@hotmail.com)

**Abstract.** Vegetation interact closely with water resources. Conventional field studies of plant-water relations are fundamental for understanding the mechanisms of how plants alter and adapt to environmental changes, while large-scale studies can be more practical for regional land use and water management towards mitigating climate change impacts. In this study, we investigated the changes in total water storage (TWS), aridity index (AI) and vegetation greenness, productivity and their interactions in the Pearl River Basin since 04/2002. Results show an overall increase trend of vegetation greenness and productivity especially in the middle reaches where TWS also increased. This region dominated by croplands was identified as the hotspot for changes and interactions between water and vegetation in the basin. Vegetation was more strongly affected by TWS than precipitation (P) at both the annual and monthly scales. Further examination showed that the influence of TWS on vegetation in dry years was stronger than wet years, while the impact of P was stronger in wet years than dry years; moreover, vegetation productivity responded slower but stronger to atmospheric dryness in dry years than wet years. The lag effects resulted in nonlinearity between water and vegetation dynamics. This study implies that vegetation in the basin uses rainwater prior to water storage until the soil gets dry, and their dynamics indicate that vegetation development is subject to water availability and that vegetation is not dominant in reducing water availability.

## 1 Introduction

Vegetation covers 70% of the land surface, playing a vital role in water, carbon and energy exchanges between land and atmosphere (Yang et al., 2016). As climate change has been more and more evident since the industrial age (Marvel et al., 2019; Sippel et al., 2020) resulting in numerous ecohydrological problems such as droughts, flooding, tree mortality, etc., managing land use especially through vegetation manipulations has been considerably practiced in many catchment planning projects (Adhami et al., 2019; Stewardson et al., 2017). The theoretical basis for vegetation-involved catchment management is the plant-water relations across multiple scales, for example, vegetation can intercept precipitation by the canopy which helps with the flood control (Soulsby et al., 2017; Wheater and Evans, 2009); they uptake soil water or groundwater and transpire it through leaves to increase moisture in the air; and the plant roots create macropores for water flow paths in soils to aid rapid recharge to soil water stores (Ghestem et al., 2011). In addition, vegetation assimilates carbon dioxide ($CO_2$) through

photosynthesis to produce food and energy materials and reduce greenhouse gas concentration (Notaro et al., 2007; Yosef et al., 2018). In turn, atmospheric and hydrologic conditions can affect vegetation growth by altering the physiological characteristics such as the openness of stomatal aperture (Reyer et al., 2013; Sala et al., 2010). Therefore, investigation of plant-water relations is of great importance in maintaining terrestrial hydrological regimes and mediating carbon cycle and energy balance in the Earth systems.

Conventional studies of plant-water relations are often carried out at the leaf and canopy level based on extensive field measurements. There are a rich pool of literatures that examine the plant responses to stress from both atmospheric conditions and water supply (Martin-StPaul et al., 2017; Whitehead, 1998). It may be true that all ecosystems are to some degree controlled by water, but the mechanisms vary greatly (Asbjornsen et al., 2011), for instance, plant water use responded sensitively to rainfall pulses and amounts in dry semi-arid areas (Huang and Zhang, 2015; Plaut et al., 2013), whilst the light exposure (i.e. radiation) between frequent low-intensity rainfall events seemed more important to stimulate transpiration than rainfall amount in the humid low-energy boreal forest (Wang et al., 2017). It is well recognized that plant-water interactions will affect soil moisture dynamics, and the soil water especially the root-zone moisture in turn plays a key role in regulating plant growth. The relationship is commonly characterized as linear increase of plant water use with increasing moisture within a certain range, above which plant water use maintains its potential rate and will be limited mainly by energy (Novák et al., 2005).

The field studies are fundamental for deep understanding of the mechanisms of how plants alter and adapt to environmental changes (Massmann et al., 2018; Petr et al., 2015; Sussmilch and McAdam, 2017). However, it is difficult to draw universal conclusions about plant-water relations extrapolative to a large landscape comprised of multiple vegetation types and with different structures from site-specific analysis (Aranda et al., 2012; Wang et al., 2008). This phenomenon is depicted as the longstanding "scale issue" in ecohydrology (Anderson et al., 2003; Jarvis and Mcnaughton, 1986), which would weaken the applicability of observation-based research outcomes during the implementation of vegetation-related ecological projects at a large scale (Liang et al., 2015). Practically, assessing and mitigating climate change impacts require effective integrated efforts at a catchment or regional scale (Fowler et al., 2019; Ma et al., 2015), therefore, it is necessary to investigate the plant-water relations at a larger scale beyond the field sites. However, data availability is often one of the greatest obstacles for large-scale and long-term ecohydrological studies. Remote sensing (RS) products are thus very useful and favourable because the abundant land surface information is beneficial especially in sparsely monitored basins in terms of overlooking the plant-water dynamics from a large area and over a long period. Over the past several decades, various RS data have been applied in many fields such as water budget assessment and hydrological components estimation (Pham-Duc et al., 2019; Wang et al., 2014a), vegetation phenological variation and the climate change impacts (Güsewell et al., 2017; Hwang et al., 2018), ecosystem services and its linkages with climate and land use (Xiao et al., 2019), etc. Vegetation dynamics can be reflected by many available indicators including reflectance-based vegetation index, leaf area index, and gross primary production (GPP). Among them, NDVI (normalized difference vegetation index) and EVI (enhanced vegetation index) as well as GPP data have been

extensively used in literature to facilitate studies of vegetation in response to climate and hydrology. For example, A et al. (2017) discussed the relationship between water storage (TWS), soil moisture and GPP in response to drought in 2011 in Texas, USA, and found that vegetation dependency on TWS weakened in the shrub-dominated west and strengthened in the grassland and forest area; Wang et al., (2020) compared phenological matrix derived by NDVI, SIF (solar-induced chlorophyll fluorescence) and VOD (vegetation optical depth) and found consistent pattern of asynchrony. The advantage of RS analysis in terms of the spatial and temporal coverage is prominent in assisting the land and water management by pinpointing the areas where the vegetation and hydroclimate changes and interactions are more sensitive.

Among the studies of plant-water relations lies an interesting and meaningful argument. On the one hand, vegetation need water to survive and thus are directly influenced by water availability. For instance, the most severe ecosystem degradation being faced by many inland river basins is closely related to reduced water availability (Yu and Wang, 2012). On the other hand, vegetation are effective conduits to return water from soils to the atmosphere through transpiration and interception loss, and thus can cause big water shortage concerns (Xia and Shao, 2008). It is found that in most cases an increase in forest cover will reduce water yield and soil water storage (Brown et al., 2005; Schwärzel et al., 2020) because of an increase in evapotranspiration, though the magnitudes are subject to scale, species and catchment size (Blaschke et al., 2008; Wang et al., 2008). Numerous studies prove that many dryland ecosystems are sourcing soil water recharged by precipitation or groundwater, therefore, plant growth depends largely on rainfall pulses or groundwater level (Eamus and Froend, 2006; Xu et al., 2016; Yang et al., 2014). While majority of such studies were carried out in semi-arid regions because of the urgent need to find an equilibrium threshold between ecological restoration and available water resources in these water-limited areas, it is still largely unclear whether the restriction of water resources or available energy on vegetation growth prevails in the humid or semi-humid areas with both abundant rainfall and radiation. The mechanisms of hydroclimate controls on vegetation can be different between arid and humid environments (Asbjornsen et al., 2011; Sohoulande Djebou et al., 2015).

In this study, we investigated the plant-water relationships in the Pearl River basin (PRB), the largest river basin in subtropical humid south China, which supports ~120 million populations. Water is one of the most important strategic resources in the basin, especially in one of its sub-basins - the East River basin. The East River basin provides water for the densely populated and highly economically developed delta region including Shenzhen and Hong Kong, and the water exploitation rate has nearly reached 38%, which increases the difficulty in water allocation and management among different administrative regions and water use sectors. Vegetation of both natural and cultivated covers vast areas of the Pearl River basin (>92%). With around half of the total annual precipitation leaving the basin as evapotranspiration (Gao, 2010), consumption of water by plants is non-negligible and may pose threats to other water cycle components like streamflow which is the major water resource in most of the basin. Despite previous studies examining the changes in vegetation greenness and investigating the roles of climate and droughts (represented primarily by temperature and precipitation) in the PRB and its sub-basins (Lin et al., 2017; Niu et al., 2018; Wu et al., 2019; Zhang et al., 2013), there are few studies quantifying how vegetation productivity alongside greenness interact with water resources from the short to long terms under contrast atmospheric dryness conditions. Such

investigation can be informative for the basin-wide land and water use planning under a rapid changing environment. Thus, the objectives of this study include (1) characterizing the spatiotemporal patterns of hydroclimate and vegetation changes in the last decade or so, identifying the hotspots for these changes and the possible driving forces; and (2) quantifying the plant-water relations at different temporal scales and under contrasting dryness conditions to determine the interactive roles of water availability and plant growth in this humid basin.

## 2 Data and Methods

### 2.1 Study area

The Pearl River (in the range of 102–116°E, 21–27°N) ranks the second largest in China in terms of streamflow with a drainage area of ~450,000 km$^2$ (Fig. 1). The climate of the Pearl River Basin (PRB) is characterized as subtropical, mainly influenced by the eastern Asian monsoon and typhoons. The long-term mean annual temperature across the basin is 14–22°C, and mean annual precipitation is 1200–2200 mm (Chen et al., 2010), decreasing from southeast to northwest and primarily falls as rain and concentrates in April-September. The elevation is ~2900 m in the west upland and decreases dramatically to the delta in the southeast, creating a maximum gradient of ~3000 m.

The dominant vegetation is evergreen forests (~65.3%), followed by cropland (~18.1%) distributed mainly in the middle of the basin along a northeast-southwest transect, where happens to be in the transitional areas of high-to-low elevations in Guangxi province. Grassland (~9.3%) is the third largest land cover type mostly located in the west upland. Due to the downstream location, flat terrain, and rapid population growth and economic development, the Pearl River Delta tends to be more and more vulnerable under natural hazards such as flood and storm surge in wet seasons and saltwater intrusion in dry seasons (Liu et al., 2019). In the recent 2 decades, droughts were found to occur frequently in the basin and affected water allocation to different municipal areas and industries (Deng et al., 2018; Xu et al., 2019).

### 2.2 Data sources and pre-processing

To assess the plant-water relations at a large spatial scale, we obtained hydroclimate and vegetation data from different sources (Table 1). Total water storage (TWS) change as one important water availability indicator is inferred by the mass change detected by GRACE satellites (Tapley et al., 2004), which can be accessed from three data processing centres. We obtained the monthly TWS anomaly (TWSA) data from Jet Propulsion Laboratory (JPL) and Center for Space Research (CSR) that are based on the 'mascons' solution (release 6) at a resolution of 0.5°. These JPL and CSR RL06 products incorporate corrections to minimize errors including the C20 coefficients corrections from satellite laser ranging (Loomis et al., 2019), the degree-1 coefficients (Geocenter) corrections, the glacial isostatic adjustment (GIA) corrections (Peltier et al., 2018), etc (Swenson and Wahr, 2006). Moreover, the CSR RL06 mascon solution uses mascon grids of 1-degree and the hexagonal tiles that span across the coastline are split into two tiles along the coastline to minimize the leakage between land and ocean signals; while the JPL RL06 mascon solution uses mascon grids of 3-degree, and leakage errors across land/ocean boundary are incorporated with

the procedure provided by Wiese et al., (2016). Save et al., (2016) reported that quantifying leakage errors does not impact

CSR mascon solutions as much as it affects JPL mascon estimate due to the native estimation resolution of 1° for CSR mascons versus 3° for JPL mascons. In addition, JPL RL06 products also provide gridded scaling factors generated by CLM land surface model to be multiplied to the mascon fields that have the CRI filter applied to calculate the final TWSA values (Landerer et al., 2020; Wiese et al., 2016). In this study, to reflect the spatial variability of the TWSA over time, we further calculated the standard deviation of the entire time series averaged over the basin, i.e. the standard deviation of TWSA in each year represents

the spatial variability of TWSA in that year across the whole basin.

In addition to water storage, precipitation (P) data were obtained from Global Land Data Assimilation System (GLDAS) (Rodell et al., 2004) and the national standard meteorological stations distributed across the basin from the China Meteorological Administration (CMA). Potential evapotranspiration (ETp) was also obtained from GLDAS and MODIS. Aridity index (AI) was calculated as the ratio of average ETp to P to represent the atmospheric dryness condition.

Vegetation data include the EVI, SIF and GPP, representing surface greenness and productivity, accordingly. EVI was obtained from the MODIS at a monthly and 0.05° resolution. SIF brings major advancements in measuring the terrestrial photosynthesis, has a strong correlation with vegetation production, and represents well the vegetation dynamics. We obtained 0.05° and monthly GOSIF data which is based on the OCO-2 datasets (Li and Xiao, 2019). Monthly GPP was obtained from three sources including MODIS (Running et al., 2004), VPM (Zhang et al., 2017b) and PML-v2 (Zhang et al., 2019).

Information of data sources for all variables is listed in Table 1. All data were resampled to 0.5° from their original resolutions. Moreover, to compare with GRACE data, anomalies of P, AI, EVI, SIF and GPP data were calculated by subtracting the means over the same baseline period of GRACE data (i.e. 01/2004–12/2009). Note that using a different baseline period such as the entire study period is also feasible, and a longer baseline period is preferable. Using a different baseline period will change the magnitude of the anomaly data series slightly but not the dynamics (i.e. fluctuation patterns and occurring time for the high

and low values) and trends which are more relevant in this study. Moreover, we checked the mean annual precipitation over the basin, and found the mean value was 1444.0±138.0 mm over the period of 2004-2009, comparable to 1461.3±150.7 mm over the entire study period. The means that the period of 2004-2009 is representative of the normal condition over the basin for this study period. All variables were obtained from 04/2002 to 03/2015 covering 13 hydrological years. Cubic spline interpolation was applied to fill the missing monthly data for the GRACE, MOD16/17 and PML.

**2.3 Associated uncertainties in the datasets**

In this study, we used remote sensing and assimilated data of water storage, vegetation status and precipitation to assess their relationships. Precipitation is one of the commonly monitored meteorological variables, usually with relatively long time series and wide spatial coverage. We compared P from GLDAS and meteorological stations in Fig. S1. It shows that the two datasets agree well both spatially and temporally. The spatial coefficients of determination ($R^2$) range from 0.7 to 0.9 in pixels where

stations are available, and the temporal $R^2$ is 0.98 with a close-to-one regression slope. The comparison indicates that the

gridded GLDAS precipitation data can be used to analyse the dynamics and relationships of hydroclimate and vegetation parameters. ETp was also compared in Fig. S2-3, which shows that spatially the correlation coefficient between monthly and annual ETp lies mostly in 0.6~1.0 and 0.4~1.0, showing relatively good agreement; and temporally ETp are close to each other at the monthly scale while the uncertainty enlarges at the annual scale.

Liu et al. (2014b) compared five GPP datasets against observations at six sites across China and concluded that MODIS GPP was more reliable over grassland, cropland and mixed forestland than the other datasets. These land cover types happen to be the predominant ones in the PRB, which assures some degree of confidence in GPP analysis using MODIS product. Zhang et al. (2017b) and Yuan et al. (2015) also compared various GPP datasets globally and regionally, and inconsistencies existed in these comparisons that could stem from the way each algorithm parameterizing atmospheric and water stress and difference

in the vegetation index data. From Fig. S4-5 for comparison of three GPP datasets, we found spatially the GPP values from MODIS and VPM are more comparable than PML which provides higher values. The annual trends inferred by the three products vary across the basin, mostly within the range of -25 to 25 $gCm^2\,yr^{-1}$. Correlation coefficients between each two GPP datasets are high at both the monthly and annual scales. It is worth mentioning that the algorithms for MODIS, VPM and PML only account for atmospheric restrictions (including vapor pressure deficit, temperature, and radiation) but barely accounts for

soil water availability (Pei et al., 2020), in which case the GPP could be overestimated. However, without extensive gridded ground observations in the basin to validate these datasets, it is hard to conclude which one is most accurate.

In lack of ground truth data, and inspired by the studies of TWS change using GRACE satellite data with different processing algorithms (Long et al., 2017; Sakumura et al., 2014), it may be more informative by using the average values from as many available datasets for the targeted variables as possible, i.e. the ensemble means, than using a single dataset. We used this

method to get the mean TWSA, ETp and GPP values for correlation analysis in this study. This may be worth further investigation which could enhance the studies in many ungauged basins for critical hydrological assessments given the increasing availability of remotely sensed and assimilated datasets.

## 2.4 Data analysis

To investigate the changes in hydroclimate and vegetation, we carried out trend analysis using the Mann-Kendall (MK) test

method both in space and in time. The MK test does not require normality of time series and is less sensitive to outliers and missing values (Pal and Al-Tabbaa, 2009). This non-parametric test method has been used in many studies to detect changing hydrological regimes (Déry and Wood, 2005; Zhang et al., 2009). Interplay between hydroclimate and vegetation was quantified by linear regression; the Pearson correlation coefficient ($r$) and coefficient of determination ($R^2$) were taken as a measure for assessment of the linkages between different variables. Data series were detrended by removing the linear trends

and deseasonalized by subtracting the multiyear monthly means when calculating the Pearson correlation coefficients. Furthermore, to investigate the interactive role of vegetation growth and water availability, we carried out lag effect analysis

between vegetation parameters and hydroclimate variables. We assume that vegetation growth is subject to water resources availability if temporal variation of vegetation parameters falls behind that of P and/or TWSA, and vice versa.

Since the interactions between hydroclimate and vegetation can be different under dry and wet conditions, we selected dry and wet years according to the national drought records as well as the annual dynamics of TWS, vegetation indices and AI under the criteria that dry conditions correspond to low negative anomaly values of TWS and GPP in addition to high positive anomaly of AI. The relationships between hydroclimate dynamics and vegetation greenness and productivity were specifically compared in these dry and wet years. Uncertainties of the data used were estimated by the standard deviation of each variable at the monthly and annual scales over the entire basin. It is worth mentioning that vegetation growth is usually controlled by two groups of factors, i.e. the demand (e.g., radiation, vapor pressure deficit, and temperature, etc) and the supply (e.g., soil moisture, groundwater, and water storage, etc). The supply control was represented by P and TWS here, and the demand effect was integrated in ETp and embedded in the aridity index. In this sense, we have the impacts of both groups accounted for on vegetation growth.

## 3 Results

### 3.1 Changes in water storage and dryness

Comparison of the GRACE data from JPL and CSR shows that mean annual TWSA from $GRACE_{JPL}$ was overall greater than that from $GRACE_{CSR}$ (Fig. 2a-b). Both products showed clear zonal characteristics similar to the average of the two (Fig. 2c) that TWSA was generally higher in the middle-to-east areas than the rest of the basin especially the west upland, which infers a generally wetting condition in comparison to the baseline period. The trends of annual TWSA (Fig. 2d) showed that over the 13 hydrological years the TWS in most of the basin has increased at a rate below 10 mm $yr^{-1}$ with 46% of the total area in the range of 5.0–10.0 mm $yr^{-1}$. Areas with low changing rate were mainly located in the west upland where the predominant land cover is grassland with underlying karst limestones. It should be noted that the spatial distribution of water storage change should be interpreted with caution as the GRACE satellites provide intrinsically ~3°-resolution coverage while the products in use are processed with different smoothing and scaling solutions to improve the spatial resolutions. It is suggested to use the spatially averaged values for temporal analysis over a large region. Nonetheless, to detect the possible hotspots for changes in water resources and vegetation, we kept the spatial analysis in this study. Temporally, the basin has been getting wetter in general from 2002 (Fig. 2e). The TWSA has increased over the 13 hydrologic years (statistically insignificant) by 6.8±2.6 mm $yr^{-1}$ inferred by $GARCE_{JPL}$ and 4.6±1.0 mm $yr^{-1}$ by $GRACE_{CSR,}$ with an average of 5.9±1.4 mm $yr^{-1}$. In the following sections, only the mean TWSA from $GRACE_{JPL}$ and $GRACE_{CSR}$ was used for analysis. Noticeably, there were three shifts in the drying and wetting tendencies over the study period, i.e. the shift from drying between 2002 and 2005 to wetting between 2005 and 2008, followed by the shift back to drying between 2008 and 2011, and finally the shift to wetting after 2011.

Fig. 3 shows the aridity index (AI, ratio of ETp to P) characterizing the spatial and temporal patterns of dryness. AI indicates the lump effect of water supply and atmospheric demand. Majority of the basin has a semi-humid climate (AI=1.0~1.5); the west upland was clearly drier than the rest of the basin which is clearly associated with precipitation patterns. Although dryness condition has not changed significantly over the 13 years with an overall positive trend spatially ($0.002\pm0.009$) and temporally ($0.005\pm0.025$), it has some interesting characteristics such as the wetting tendencies primarily located in the cropland areas, and the alternate periodical wetting and drying episodes temporally also existed like TWSA. Areas with low TWS change rates generally coincided with drying climate represented by aridity index. Combining TWSA and AI dynamics, we were able to define the relatively dry and wet years during the study period.

## 3.2 Changes in vegetation greenness and productivity

Spatial EVI distributions (Fig. 4A) were highly related to vegetation cover types that the high EVI values coincided with forest covers and low values corresponded to impervious surfaces, grasslands and croplands. It clearly reflects the impacts of urbanization on surface greenness particularly near the basin outlets in the southeast. Over the 13 years EVI has shown significant increases across the basin, and the majority (~78.7%) had a MK test $p<0.05$ at the pixel level. The areas with greater increases were mostly concentrated in the central south of the basin where croplands are predominant, indicating a possible intensification of crop farming activities over these areas. Temporally, EVI has an overall significant increase trend over the 13 years at an annual rate of $0.004\pm0.003$ ($p<0.001$). It is noticeable that the periodical shifts in the EVI trends were just slightly different to TWSA in Fig. 2e. This reflects a tight bound between the vegetation greenness and water availability in this rain-abundant region at the annual scale. Interestingly, in 2004-2005 when water storage continued to decrease following the previous years, EVI did not show a continuity of decreasing but increased instead, coincided with a slight decrease in aridity index.

As an effective probe for photosynthesis, SIF showed almost identical patterns and trends with EVI (Fig. 4B), i.e. high values distributed in forests and low values in croplands and grassland. Over the years SIF has increased significantly by $0.003\pm0.012$ W m$^{-2}$ μm$^{-1}$ sr$^{-1}$ per year ($p<0.001$). Another vegetation biomass parameter GPP was also analysed for the basin (Fig. 4C). It is not surprising to observe that GPP was highly responsive to EVI and SIF such that areas with low EVI and SIF also had low GPP (e.g., the central agricultural region and upland grassland). GPP anomaly also showed positive high values in the central south areas dominated by croplands coincident with EVI and SIF anomaly. It should be noted that the trends were statistically significant only in 33.6% of all pixels, many of which are located in the cropland areas. Over the entire basin, annual GPP showed almost the same periodical decreasing and increasing trends as EVI and SIF, except that the first turning point occurred in 2005 rather than 2004. Linear regression gave a coefficient of determination $R^2$=0.44 ($p$=0.014) between annual TWSA and EVI, $R^2$=0.41 ($p$=0.019) between TWSA and SIF, both higher than that between TWSA and GPP ($R^2$=0.23, $p$=0.099), which may imply a more direct and stronger dependence of vegetation greenness than productivity on water storage at an annual scale.

**3.3 Interactions between hydroclimate and vegetation**

Combining Fig. 2-4, we found that climate condition, water storage and vegetation dynamics are tightly interlinked. Coefficient of determination between anomalies of these variables (Fig. 5) show that variation of annual EVI can be explained by TWS by 43.7% ($p$=0.014), followed by P (14.7%, $p$=0.196) and AI (4.6%, $p$=0.479). Influence of these three variables on GPP and SIF followed the same order ($R^2$=0.23, 0.06, 0.02 for GPP, and $R^2$=0.41, 0.12, 0.03 for SIF) but not statistically significant ($p$>0.05, except for TWS and SIF). In addition, GPP was positively associated with EVI and SIF ($R^2$=0.53, $p$=0.005; $R^2$=0.51,

$p$=0.006). SIF and EVI are highest correlated among the three vegetation parameters with a $R^2$=0.95 ($p$<0.001). P and TWS were negatively correlated with dryness ($p$<0.05).

Spatially, precipitation, water storage and dryness affected vegetation in a similar way compared to temporal characteristics, i.e. the influence of TWS was relatively stronger than P and AI. The hotspots of the interactions were found in the middle-south areas, and dryness more negatively affected greenness than productivity in these areas (Fig. 6). Atmospheric stress and

water stress imposed more direct and stronger impact on vegetation greenness than productivity on a yearly basis, and water constraint on vegetation was stronger than that of dryness. It should be noted again that due to the intrinsic resolution of GRACE satellite imaging the pixelwise calculation of correlation coefficient between TWSA and other parameters cannot necessarily accurately represent their quantitative relationships, and therefore, should be interpreted with caution. Here we kept the figure with the intention to find the possible hotspots of intensive interactions.

At the monthly scale, over the entire basin, the linear responses of GPP to P and TWS were slightly weaker than the linear responses of EVI and SIF to P and TWS (Fig. 7a-b). The response of both EVI, SIF and GPP to P was more nonlinear than to TWS, and the sensitivity of EVI, SIF and GPP to TWS was all stronger than to P indicated by the linear regression slopes, implying a stronger link between vegetation growth and water storage than precipitation. Meanwhile, increase in dryness resulted in strong nonlinear decreases in all vegetation parameters (Fig. 7c). The relationships show that although precipitation

is the main water input to the terrestrial hydrological cycle, it is how much water is stored in the soils that determines vegetation greenness and biomass production, and the atmospheric constraints on vegetation is more complex than water supply. Nonlinear plant-water relationships can be explained by the lag effect that monthly changes of EVI, SIF and GPP fell behind the changes of P and TWS to varying degrees (Fig. 8). Based on our assumption, it is more likely that the decline of water resources deteriorates vegetation greenness and productivity, not the opposite. This means that the leading role of water

availability on vegetation growth outweighed the impacts of vegetation growth on water resources reduction.

Vegetation response to hydroclimate changes is expected to differ in dry and wet years. Here, we assumed that the annual anomalies of TWS<0, EVI<0 and AI>0 corresponded to dry conditions, and hence defined 2003, 2005, 2007, 2009 and 2011 as dry years and 2002, 2006, 2008, 2010, 2012-2014 as relatively wet years. There was evidence of drought occurrences in these dry years (Lin et al., 2017; Wang et al., 2014b). It can be seen that the dry and wet years were mainly differentiated by

the rainfall data in spring-summer months, resulting in obviously lower water storage and higher dryness in autumn-winter in dry years than wet years. Noticeably, although the hydroclimate conditions differed greatly the vegetation parameters showed

similar patterns and ranges (Fig. 8c-d). While the maximum and minimum GPP was slightly higher (14.9%) and lower (14.3%) in dry years than wet years, respectively, the EVI and SIF did not show such distinct differences (<5%). This implies that vegetation greenness is less sensitive to any changes in hydroclimate than productivity, and that GPP during dry periods was relatively higher than that in wet periods, reflecting a positive effect of water stress on biomass production, but this could be mainly attributed to anthropogenic intervention such as probable water surplus via irrigation during dry periods.

Fig. 9 gives the $R^2$ from linear regression between the monthly climatological means of different variables considering phase shift for lag analysis over all the years, dry and wet years, respectively. It shows EVI, SIF and GPP varied strongest with P, TWSA and AI in the previous 1, 0 and 1 month, respectively. Comparison of the lag time in dry and wet years shows that the influence of P on vegetation was more prominent in wet years than in dry years, while TWS influence was greater in dry years than wet years. There was high consistency between vegetation change and water storage change with zero lags, in comparison to 1-month lag between vegetation parameters and precipitation. Moreover, response of GPP and SIF to dryness change was 1 month slower in dry years than wet years.

## 4 Discussion

### 4.1 Hotspot for hydroclimate and vegetation changes

The three investigated vegetation parameters shared the same spatial patterns and high GPP corresponded to high EVI and SIF in the forested areas; low values existed in the west upland with grass cover and the central south areas of croplands. Over the 13 hydrologic years EVI and GPP have increased significantly by 0.004 (unitless, $p<0.001$) and 8.57 gCm$^2$ yr$^{-1}$ ($p=0.038$), respectively, with periodic decreasing and increasing tendencies. Unlike the north China where vegetation cover is deeply affected and largely recovered through decades of ecological restoration projects (Chen et al., 2019; Feng et al., 2005), vegetation cover especially the forest cover which occupies most of the PRB remained near-constant from early 2000s at least in Guangdong province located in the east of the basin (Chen et al., 2015). We identified the areas with significant increase in the vegetation parameters in the central south region of the basin where croplands dominate. The changes of TWS, EVI, SIF and GPP jointly imply that the water storage increase in this hotspot region, which was likely induced by increased precipitation, coincides with the intensification of agricultural activities and boosted the food production since the early 2000s. That is, the intensive vegetation greening most likely was not only induced by natural hydroclimatic changes but also intervened by agricultural activities such as planting structure adjustment and irrigation during dry spells. Tong et al., (2018) showed that vegetation greenness and aboveground biomass production have increased in southwest China, in spite of drought conditions. The increases have been attributed largely to land use type conservation mainly through ecological engineering such as reforestation, etc. Their study area partly overlapped with this study, and their results support our speculation indirectly that the agricultural activities in this cultivated area have intensified and thus enhanced vegetation growth. Changes in planting structure in these agricultural areas could also result in enhanced greenness and improved productivity compared to the traditional cultivated crops, but this cannot be quantified without detailed crop data throughout the years. Nonetheless, it is for

the first time in studies to reveal such phenomenon and can be meaningful for the food-water management dies in this region, and indicative for a possible expansion of China's main food production from the north to the south in the context of water and energy richness in the south and shortages in the north (Kuang et al., 2015).

## 4.2 Interactive roles of water supply, demand and vegetation changes

The overall TWS increase is promising for the managers and users of water resources in the PRB, however, the strong correlation with precipitation seasonality restrained the available water in the relatively dry periods which would raise concerns on water shortage under drought conditions. In fact, previous studies have reported the contribution and restriction of P to TWS. For instance, Chen et al. (2017) revealed the liability of P to TWS ($r$=0.78) in the PRB. Mo et al. (2016) found TWS more strongly explained (60%) by annual P in river basins in south China than in north China. In this sense, storage shortage in dry periods subject to seasonal reduction of precipitation would hamper vegetation greening. Analysis in this study shows that EVI was highly correlated with TWS and P at the annual scale, consistent with previous studies in the PRB and other areas (Guan et al., 2015; Zhaos et al., 2016; Zhu et al., 2018). Whilst at the monthly scale EVI, SIF and GPP were all strongly associated with TWS but slightly less strongly with P. The differences in EVI and GPP response to hydroclimate variables may lie in the way these two parameters are calculated, especially that GPP is formulated by atmospheric variables like temperature, vapor pressure deficit and photosynthetically active radiation (Pei et al., 2020). Because of the asynchrony in the atmospheric variables and vegetation greenness (Kong et al., 2020; Piao et al., 2006), the GPP and EVI would also have some inconsistency in time. This would then further indicate that it should be given more caution when choosing parameter (EVI, NDVI, SIF or GPP) to better represent vegetation phenological features, which is still lack in literature for discussion (Kong et al., 2020; Wang et al., 2020). The weakened linear influence of P on vegetation parameters at the monthly scale, found also by others such as Bai et al. (2019) and A et al. (2017), can be explained by the lag effect that EVI, SIF and GPP lagged by 1 and 0 months after P and TWS, respectively.

Comparison of the plant-water relations in dry and wet years showed a slower response of GPP to aridity index in dry years than wet years. Wilcoxon rank sum test shows that the areal mean EVI and GPP in dry years are not significantly different from those in wet years. In fact, GPP was slightly higher in the growing seasons in dry years than wet years. These comparisons may imply that a certain degree of drying can stimulate biomass production. This phenomenon is also revealed by other studies (Zhang and Zhang, 2019). The underlying mechanisms could be similar to the principle of regulated irrigation in agricultural practice to increase water use efficiency under a certain degree of water stress (Chai et al., 2016), or that the atmospheric conditions are more favourable for photosynthesis during dry years than wet years (Restrepo-Coupe et al., 2013; Zhang and Zhang, 2019), provided that the soil water or groundwater storage is not depleted severely in these dry years. This dryness effect on ecosystem productivity cannot be detected in the annual scale assessment (Brookshire and Weaver, 2015; Yao et al., 2020). These results imply that pre-growing season hydroclimate conditions play a key role in the follow-on vegetation growth and production (Kong et al., 2020; Piao et al., 2006; Wang et al., 2019), and that vegetation dynamics are subject to not only atmospheric changes but also water resource availability even in this humid subtropical radiation- and rain-abundant region,

while vegetation development is not dominant in reducing water availability. The causal role of vegetation in water decline has been reported at mostly a shorter time scale like daily and sub-daily, such as the studies in a poplar stand in Northwest China (Shen et al., 2015), a pine-dominated catchment in Sierra Nevada, USA (Kirchner et al., 2020), and a mixed forest in the Czech Republic (Deutscher et al., 2016), who demonstrated that sap flow by trees led to decline in groundwater level and streamflow.

The drying episodes confined the vegetation greenness and production. Liu et al. (2014a) reported that China's national total annual net ecosystem productivity exhibited declines during 2000-2011, mainly due to the reduction in GPP caused by extensive drought. Although drought is generally associated with declines in vegetation greenness and productivity due to water and heat stresses (Eamus et al., 2013), the magnitude of vegetation reduction, determined by ecosystem sensitivity to drought, can vary dramatically across plant communities and thus show different spatial patterns relative to different vegetation types. While Zhang et al. (2017a) detected insensitivity of vegetation to droughts in humid south China including the lower reach of PRB, this study observed that EVI experienced a recovery in 2004-2005 after drought in the previous year, which may be a result of irrigation during drought in the agricultural regions since forests are more resilient to droughts (DeSoto et al., 2020; Fang and Zhang, 2019). Future climate projections predict increases in temperature and insignificant changes in precipitation in the basin which would trigger more heatwave induced flash droughts (Li et al., 2020). This would likely enhance the atmospheric controls on vegetation development. To mitigate the impacts on both water resources and ecosystems, proper plans should be made such as conversion of the low resilient ecosystems to forests (Fang and Zhang, 2019; Tong et al., 2018) and improvement of biodiversity in ecosystems (Isbell et al., 2015; Oliver et al., 2015), in addition to engineering regulations like reservoir operations (Lin et al., 2017).

**5 Conclusions**

Plant-water relations over the Pearl River Basin were examined using remote sensing products during the hydrological years of 2002-2014. Results show that water storage has increased across the entire basin at an average rate of 5.9 mm yr$^{-1}$. Vegetation greenness and productivity has also shown some changes with significant increases concentrated in the cultivated lands. Spatial characterization reveals that the central south areas of the basin dominated by croplands are the hotspots for the changes of and interactions between hydroclimate and vegetation. This implies an increase in vegetation is not only a result of natural hydroclimatic controls but also anthropogenic interventions such as to secure food production by intensification of agricultural activities in these areas. Lag effect analysis at the monthly scale reflects that even in this rain-abundant subtropical basin the water supply dominance on vegetation dynamics precedes the water resource reduction by vegetation development. Furthermore, a stronger influence of precipitation and a weaker influence of water storage on vegetation were found in wet years than dry years. A slower response of vegetation productivity to aridity index in dry years than wet years was identified which may indicate a stimulating role of a certain degree of drying on vegetation production. Therefore, essentially the vegetation growth in this subtropical humid region is more strongly controlled by atmospheric demand factors than water

supply factors at the monthly scale. This study reveals the changes and interplay between plant and water using readily

available remote sensing and assimilated data and has implications for proper measures regarding land use alterations to mitigating frequent drought impacts on water resources and ecosystems under a warming climate.

**Data availability**

The original data in the study are freely available from the links given in Table 1.

**Author contribution**

Wang: Conceptualization, Methodology, Writing – original draft, review& editing; Duan: Methodology, Writing – review & editing; Liu & Chen: Writing – review & editing, Validation.

**Competing interests**

The authors declare that they have no conflict of interest.

**Acknowledgements**

This work is supported by the Guangdong Provincial Department of Science and Technology, China (2019ZT08G090), and the Open Research Fund of State Key Laboratory of Simulation and Regulation of Water Cycle in River Basin, China (IWHR-SKL-201920), and the National Natural Science Foundation of China, China (51909285). We thank Prof. Qiang Zhang and two anonymous reviewers for their invaluable comments and suggestions to improve the manuscript.

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

**Table 1.** Information of data used in this study

| Variable | Product | Resolution | Time span | Data link |
|---|---|---|---|---|
| **P** | GLDAS-Noah (v2.1) | 0.25°×0.25°, Monthly | 04/2002–03/2015 | https://disc.gsfc.nasa.gov |
| | CMA | Station-based, monthly | 04/2002–03/2015 | http://data.cma.cn/data |
| **ETp** | GLDAS-Noah (v2.1) | 0.25°×0.25°, Monthly | 04/2002–03/2015 | https://disc.gsfc.nasa.gov |
| | MOD16A2 | 0.05°×0.05°, Monthly | 04/2002–12/2014 | http://files.ntsg.umt.edu/data/NTSG_Products/MOD16/ |
| **TWSA** | GRACE$_{JPL}$, GRACE$_{CSR}$ (RL06) | 0.5°×0.5°, Monthly | 04/2002–03/2015 | http://grace.jpl.nasa.gov; www2.csr.utexas.edu/grace/RL06_mascons.html |
| **EVI** | MOD13C2 | 0.05°×0.05°, Monthly | 04/2002–12/2014 | https://lpdaac.usgs.gov/products/mod13c2v006/ |
| **GPP** | MOD17A2 | 0.05°×0.05°, Monthly | 04/2002–12/2014 | www.ntsg.umt.edu/project/modis/mod17.php |
| | VPM | 0.05°, monthly | 04/2002–03/2015 | https://figshare.com/articles/Monthly_GPP_at_0_05_degree/5048113 |
| | PML-v2 | 0.05°, 8-day | 07/2002–03/2015 | https://github.com/kongdd/PML |
| **SIF** | GOSIF-v2 | 0.05°, monthly | 04/2002–03/2015 | http://data.globalecology.unh.edu/data/GOSIF_v2 |


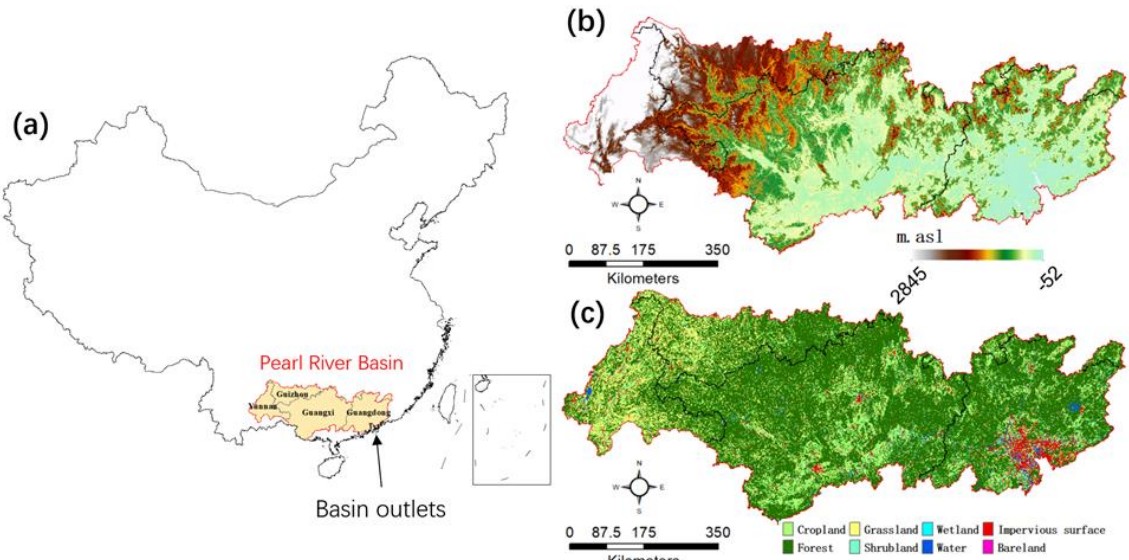

**Figure 1**. (*a*) The Pearl River Basin and the related provinces on the map of the China, (*b*) Digital elevation map (m.a.s.l, 1000 m resolution), and (*c*) Land cover types (30 m resolution).


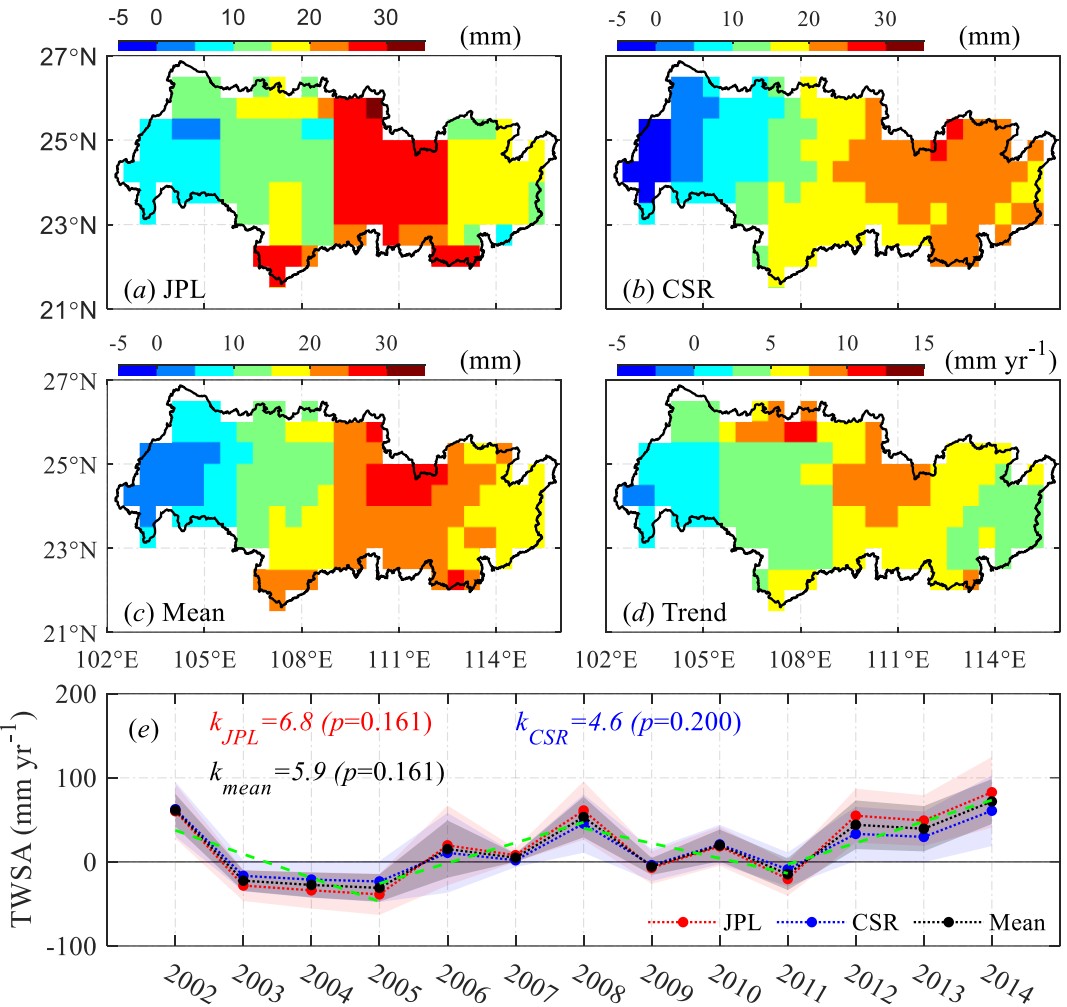

**Figure 2.** Spatial distribution of TWSA in the basin inferred by (*a*) GRACE_JPL, (*b*) GRACE_CSR, (*c*) the mean of GRACE_JPL and GRACE_CSR, (*d*) the linear trends of the mean annual TWSA, and (e) mean annual TWSA over the entire basin. Shaded areas in (*e*) show the standard error of each series. Dashed green lines indicate statistically insignificant trends ($R^2$=0.68, 0.82, 0.58 and 0.83, respectively).


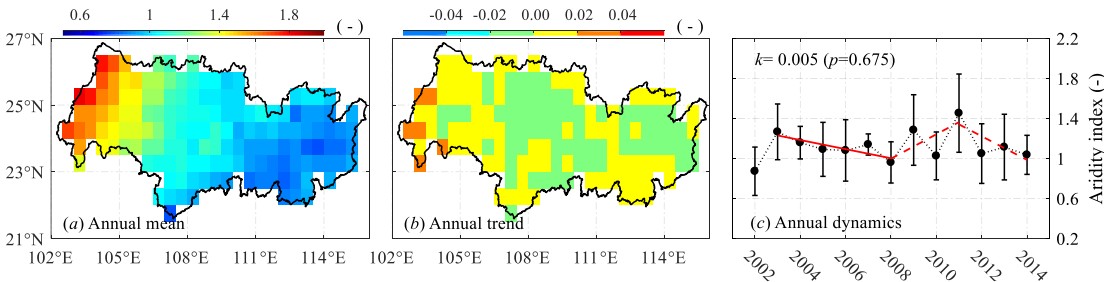

**Figure 3**. (*a*) Spatial distribution of the mean annual aridity index across the basin during hydrological years 2002-2014, (*b*) annual trend of aridity index, and (*c*) mean annual aridity index over the basin. Red lines show the periodical trends. Dashed red line indicates statistically insignificant trend. The coefficient of determination is 0.71, 0.47 and 0.61, respectively.


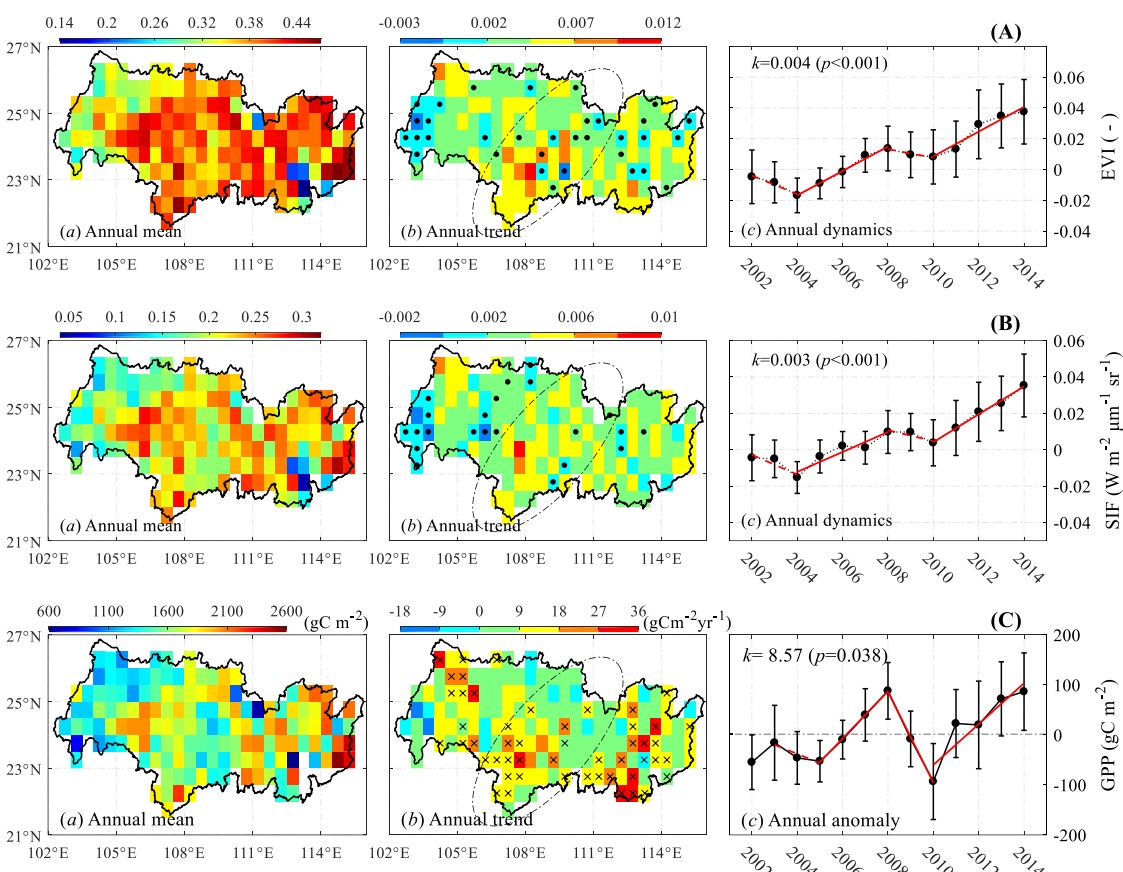

**Figure 4**. Spatial distribution of mean annual values, linear trends and temporal dynamics for (A) enhanced vegetation index (EVI), (B) solar-induced chlorophyll fluorescence (SIF); and (C) gross primary production (GPP), during hydrologic years 2002-2014. Red lines show the annual trends in different periods. Dashed red lines show statistically insignificant trends ($p>0.05$). Ellipse marks the areas where croplands predominate. Black dots indicate $p>0.05$ for the trend in the relevant pixels, and black crosses indicate $p<0.05$ for the trend in the relevant pixels.


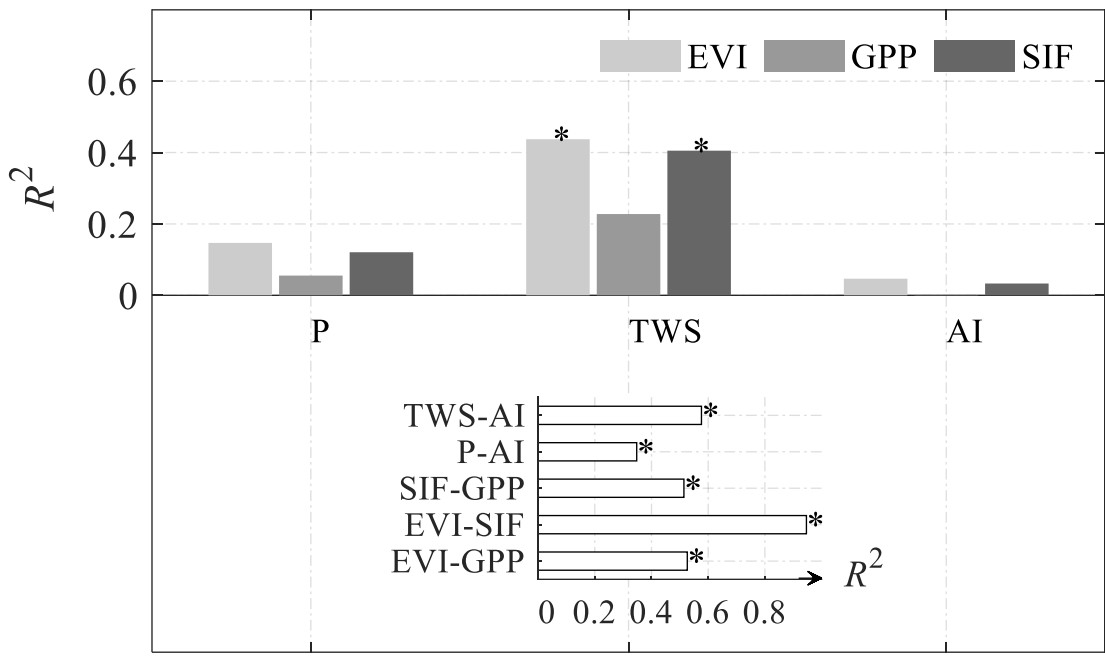

**Figure 5**. Coefficient of determination ($R^2$) from linear regressions between the basin-averaged anomalies of P, TWS, AI, EVI, SIF and GPP at the annual scale. Asterisk indicates $p<0.05$.

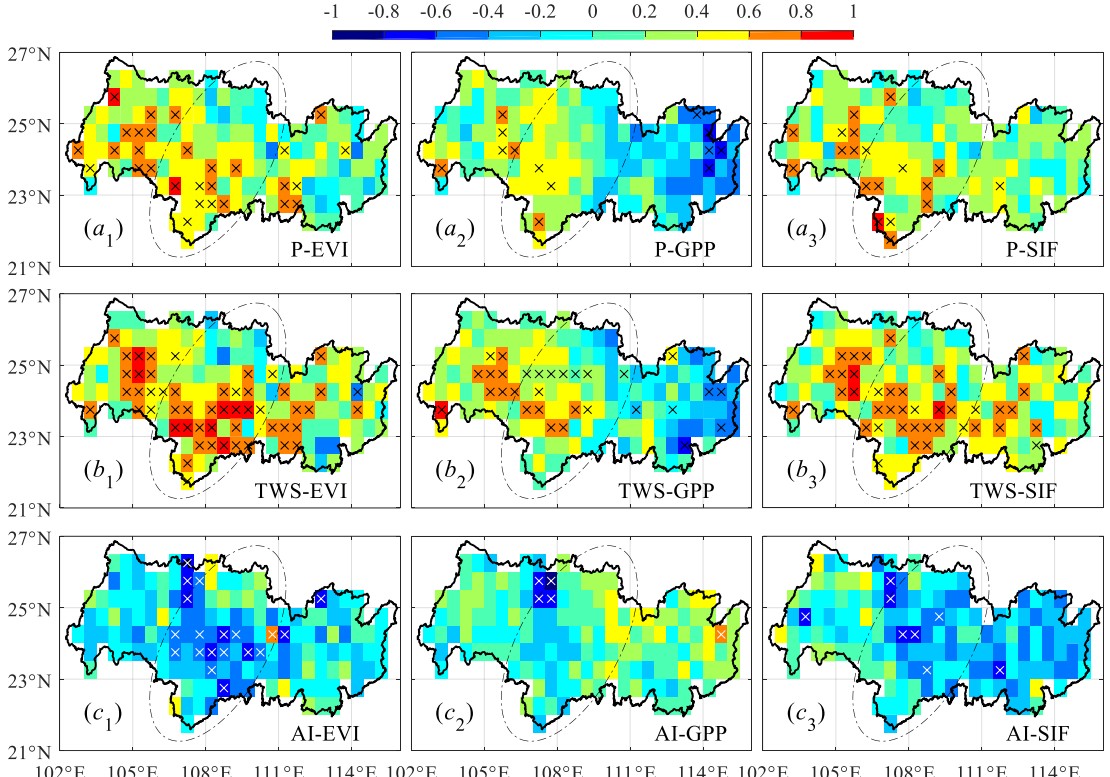


**Figure 6**. Pearson correlation coefficient between annual anomalies of ($a_1$-$a_3$) precipitation, EVI, GPP and SIF; ($b_1$-$b_3$) total water storage, EVI, GPP and SIF; and ($c_1$-$c_2$) aridity index, EVI, GPP and SIF. Ellipse marks the areas where croplands predominate. Crosses (black and white) indicate $p<0.05$.

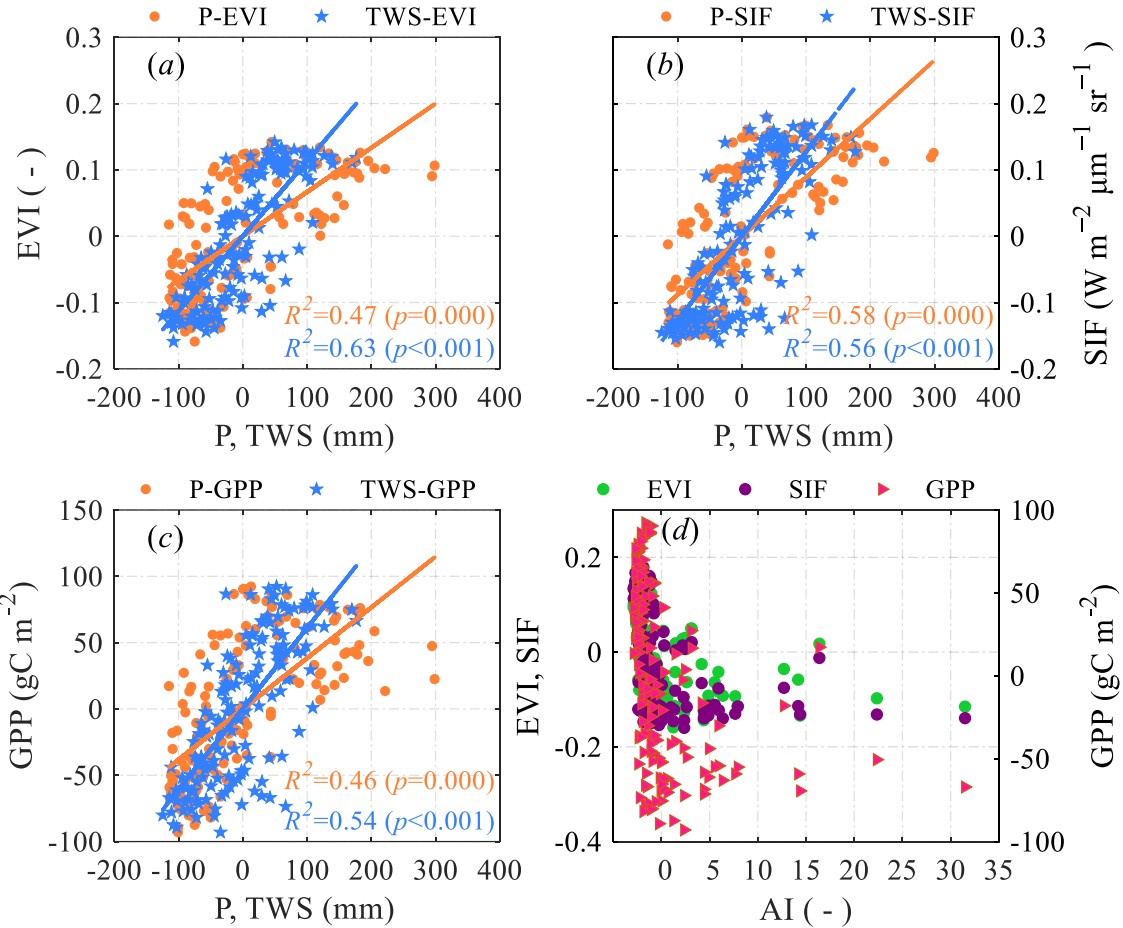


**Figure 7**. Scatter plot of monthly anomalies of precipitation (P), total water storage (TWS), aridity index (AI), enhanced vegetation index (EVI), solar-induced chlorophyll fluorescence (SIF) and gross primary production (GPP) over the entire basin. Each point represents a basin-wide averaged monthly data.

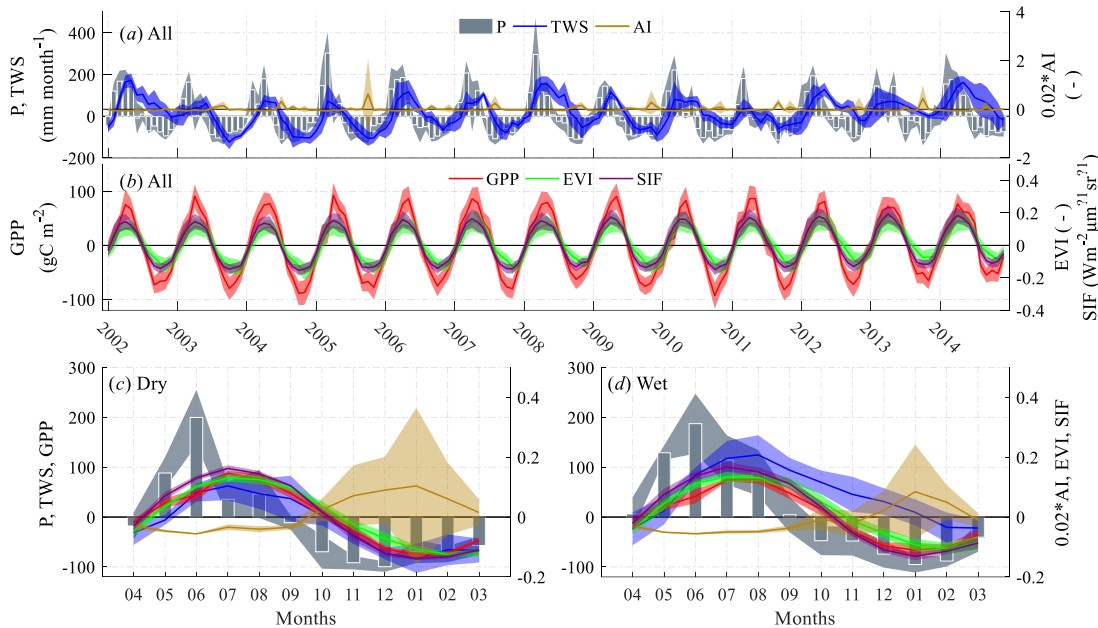


**Figure 8**. (*a-b*) Monthly variations of anomalies of precipitation (P), total water storage (TWS), aridity index (AI, scaled for a better view), enhance vegetation index (EVI), gross primary production (GPP), and solar-induced chlorophyll fluorescence (SIF) in all years; (*c*) monthly climatological means of the variables in dry hydrological years and (*d*) monthly climatological means in wet hydrological years during 2002-2014. Plots *c* and *d* share the same units and legends with plots *a* and *b*.
Shaded areas show the standard errors of each variable representing the spatial variability.

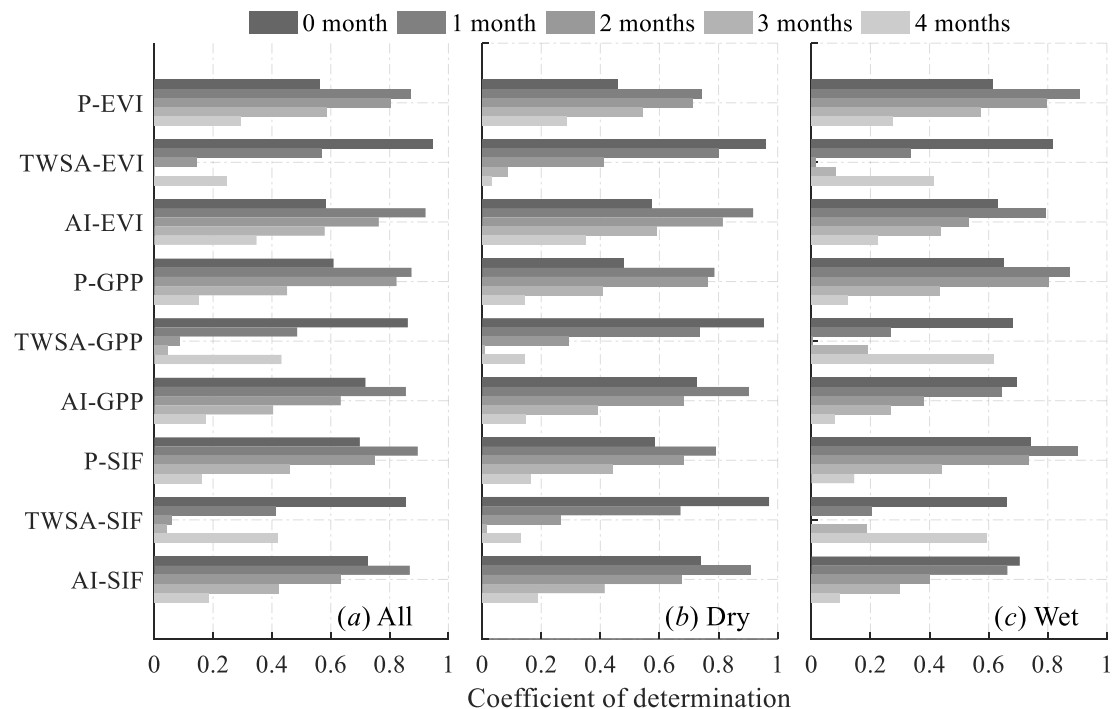

**Figure 9.** Coefficient of determination between monthly climatological means of the anomalies of precipitation (P), total water storage (TWS), aridity index (AI), enhance vegetation index (EVI), solar-induced chlorophyll fluorescence (SIF) and gross primary production (GPP) in (*a*) all years, (*b*) the dry years, and (*c*) the wet years after shifting different number of months as indicated in the legend.