# Peer review of "Assessing the large-scale plant-water relations in the humid subtropical Pearl River Basin of China"

_Hydrology and Earth System Sciences, 2020_

## Short Comment (SC1) · 17 Jul 2020

Review for hess-2020-242 Assessing the large-scale plant-water relations using remote sensing products in the humid subtropical Pearl River Basin in south China by Wang et al. This study focuses on the Pearl River Basin in south China. Water is one of the critical resources to sustain the rapid socioeconomic development. Vegetation cover is high and most of them is evergreen subtropical species, which means there could be substantial water consumption by plants throughout the year given the favorable climate. Water and ecological management faces increasing challenges because of the rapid population growth, high urbanization and industrialization, etc. Therefore,

this is a timely study to investigate the intensive changes and interactions between vegetation and water. The most interesting part of this study, to me, is the identification of the 'hotspot' of changes and determination of water-limiting-vegetation even in this rainfall-abundant region. The m/s is well structured, the wordings are fine, and method/analysis are appropriate. In my view, this study is meaningful and worth publishing.

Having said these, I do still have a few concerns listed below for the authors to address: 1. I find the summary of the water-plant relationships in Line 61-69 is quite interesting. Vegetation consumes water and causes reduction in water resources on the one hand, and water availability will restrict vegetation establishment and growth on the other. Indeed, plant-water relations are examined mostly in arid and semi-arid regions for the purpose of water and ecological conservation. Are there such studies in humid and semi-humid regions investigating the controlling factors – energy vs. water - of vegetation growth? It is important and is the authors responsibility to ensure a thorough literature review on this subject. 2. A brief paragraph should be added before Line 74 for an introduction of relevant studies that have been carried out in the Pearl River Basin. Without this, it is a bit out of blue to see the next paragraph suddenly mentioning something in this basin. 3. Regarding the data: I see a comparison between GLDAS precipitation and the ground truth data over a number of pixels given in Fig. 11. GRACE data from different processing centers are also compared. No comparisons/discussions are given for ETp and other variables. Can you find some studies in this basin or a basin with similar vegetation cover and climate that use GPP from MODIS? If there is any, it'd provide more confidence in the results of this study. 4. The current m/s is a complete story by overlooking the water-vegetation relationships in the entire basin in space and time. It is good to locate the hotspots of changes and interactions because these areas would usually be the 'focus' of land/water management and for risk control, etc. I recommend the authors to take a further step to investigate the reasons behind the changes and interactions right in these hotspot areas. 5. Paragraph ends with Line 279: This is a good argument that vegetation relies on water because of

the lags of vegetation parameters after water input & storage dynamic change. However, there seems a lack of support to the opposite standing, i.e. vegetation growth does not result in excessive water reduction. So this part of discussion needs a further expansion. 6. Fig. 2-5, 7: the spatial distributions of these variables/trends are shown for all pixels. How would it be like if only the ones with $p < 0.05$ are shown?

---

## Referee Comment (RC1) · Anonymous Referee #1 · 13 Aug 2020

The manuscript evaluates regional scale plant-water relations in the Pearl River Basin. The authors find a strong inter-annual correspondence between NDVI and GRACE-derived TWS, suggesting water limitation in an area where rainfall is generally higher than the potential evapotranspiration. This is an interesting result, but the underlying mechanism remains unclear.

The introduction touched on a few important topics such as water limitation and plant water use, but the scientific hypothesis/questions are not clearly defined. "Quantifying the plant-water relations at different temporal scales under different dryness conditions" is a good starting point, but the specific questions to address need to be defined.

[Figure]

The choice of vegetation data needs justification. NDVI is known to saturate in the forest ecosystem. MODIS GPP poorly represents soil moisture limitation on productivity which is directly relevant to the main theme of this study. There are many other vegetation metrics available that are not or less affected by these issues (e.g., SIF and EVI). LAI has also been used in a similar domain (Tong et al., 2018). I suggest the authors adopt these other datasets in the analysis.

The strong inter-annual correspondence between NDVI and TWS is interesting, given how humid this area is. It would be of interest to see if this correspondence changes across different biomes (e.g. crops vs. forests) or regions with different levels of aridity, which may be done at mascon resolution. On the other hand, the monthly-scale correlation analysis needs clarification. Is the trend and seasonality removed from the monthly time series?

The discussion session lacks a clear focus and sometimes reads like a literature review (e.g. Line 280-294). The discussion should be centered on clearly defined research questions and based directly on the results of this study.

Detailed comments:

Lines 72-73. This statement needs clarification. Is it to question if water limitation prevails in the humid ecosystems in the long term?

Line 105. I think it is better to define the TWS anomaly using the entire analyzed period as a baseline (by removing the mean calculated over the entire period), unless there are specific reasons to believe that the 2004-2009 period better represents a "normal" condition.

Lines 129-132. The mean annual TWSA depends on the choice of the reference period. The trend analysis is a better way to illustrate wetting/drying information. Are all the trends significant in Fig 2d?

Fig 2e. Please clarify how the basin average and the associated errors (measurement

and leakage) are calculated. This should be included in the Method session.

Line 145. What is the trend in space? Note that here the trend in time does not have an error bar.

Figs 3-5. Please change the color scheme to improve the readability of the figures. For example, a sequential colormap is ideal for the aridity index. For the anomaly and trends, it is better to use a diverging colormap with a symmetric scale.

Line 153. Please label the significant trends in the map.

Lines 159-161. This reads like discussion, not actual results.

Lines 169-170. Needs other proxies for plant productivity to confirm this. MODIS GPP directly accounts for the limitation from VPD but not from soil moisture supply.

Fig 7. Please either label the areas with significant correlations or mask the insignificant ones. Trends can inflate the correlation results. Have you de-trended the time series?

Line 182. It is unclear how the monthly scale regression is calculated. Note that to quantify water limitation, the seasonality should be removed from the monthly time series.

Lines 189-190. It is unclear what this means. How are the water restriction and water consumption quantified and compared? In fact, quantifying the amount and timing of plant water consumption (e.g. ET in wet and dry years) might be helpful to understand why there is an apparent water restriction in such a humid area.

Line 196. How is the span of the growing season defined in this area?

Lines 212-220. This should go to the Data and Method session.

Line 230. The uncertainty of the trend needs to be evaluated.

Lines 232-241. This should go to the Data and Method session. The authors present examples where MODIS GPP shows consistency with other vegetation data, but in this

study, the analysis based on the two datasets (MODIS GPP and NDVI) shows different plant-water relations. It is unclear if the difference is physical (e.g. due to the different responses of vegetation state and vegetation productivity) or caused by data accuracy issues. In this case, other vegetation metrics are needed to justify the results.

Line 257. Note that this is an active area for ecological restoration, including the Grain to Green project (Tong et al., 2018).

Lines 272-275. This point seems important but is not fully developed. Are there results in this study showing enhanced or perhaps near-normal productivity under drier than normal condition?

Reference: Tong, X., Brandt, M., Yue, Y., Horion, S., Wang, K., Keersmaecker, W. De, . . . Fensholt, R. (2018). Increased vegetation growth and carbon stock in China karst via ecological engineering. Nature Sustainability, 1(1), 44–50. https://doi.org/10.1038/s41893-017-0004-x

---

## Referee Comment (RC2) · Anonymous Referee #2 · 19 Aug 2020

Wang et al. (2020) explored basin-scale changes in total water storage (TWS), aridity index (AI) and vegetation greenness, productivity and how plants interact with water resource in the Pearl River Basin, China using remote sensing observations. The paper is well-organized and findings from this study are valuable for improving our understanding about large spatial scale vegetation-water relations in subtropical regions. The paper deserves publication after addressing the following comments.

General comments:

1. The Pearl River Basin is in relatively humid region. Beside water, other factors may also influence the vegetation growth. It is suggested to show the landcover change in

the studied period and analyze the relationship between vegetation growth and temperature or egergy to identify the vegetation-water relation more clearly.

2. Lag effect between vegetation growth and water availability are analyzed at monthly scale. In my opinion, it is necessary to show how P, TWS, NDVI and GPP for the 12 months in a year for better discussion about the lag effect.

3. I understand when using remote sensing products, uncertainty issue is always a concern need to be addressed. However, this is not the scientific target of this paper. To keep the readers' attention to the key scientific question trying to answer, it is suggested to remove the "uncertainties in the datasets and results" section and describe how you quantify the uncertainty of remote sensing data in the Methodology section.

Specific comments:

Line 77: Please give more information about the importance of Pearl River Basin and it's connection with research progress described in the previous paragraph.

Line 120: It is suggested to decide the assumption being made behind the lag effect analysis

Line 195: The basin is in subtropical region. So please confirm whether October to March is non-growing seasons.

Line 252-253: A landcover change analysis for the study period may make the explanation here more persuasive.

Line 254: I'm a little bit confused about "water storage increase in this hotspot region has resulted in the intensification of agricultural activities". More explanation is needed.

Figure 9: It is hard to read as many elements are overlapped together. Please find a clearer way to describe the information contained in this figure.
* * *
242, 2020.

---

## Author Comment (AC1) · 17 Sep 2020

SC1: Water is one of the critical resources to sustain the rapid socioeconomic development. Vegetation cover is high and most of them is evergreen subtropical species, which means there could be substantial water consumption by plants throughout the year given the favourable climate. Water and ecological management faces increasing challenges because of the rapid population growth, high urbanization and industrialization, etc. Therefore, this is a timely study to investigate the intensive changes and interactions between vegetation and water. The most interesting part of this study, to me, is the identification of the 'hotspot' of changes and determination

of water-limiting-vegetation even in this rainfall-abundant region. The m/s is well structured, the wordings are fine, and method/analysis are appropriate. In my view, this study is meaningful and worth publishing. Having said these, I do still have a few concerns listed below for the authors to address: AC: We thank Prof. Zhang for the encouraging comments on our manuscript, and we are glad to know that the main points of our manuscript were recognized. SC1: 1. I find the summary of the water-plant relationships in Line 61-69 is quite interesting. Vegetation consumes water and causes reduction in water resources on the one hand, and water availability will restrict vegetation establishment and growth on the other. Indeed, plant-water relations are examined mostly in arid and semi-arid regions for the purpose of water and ecological conservation. Are there such studies in humid and semi-humid regions investigating the controlling factors – energy vs. water - of vegetation growth? It is important and is the authors responsibility to ensure a thorough literature review on this subject. AC: The possibility of different vegetation-water relationships under contrast climate conditions is the motivation of this study. Since most such studies focus on drylands because of the water scarcity, we would like to know what the relationship is like in wet/humid areas. Although there have been studies in humid areas investigating environmental controls on plant water use (as you may find in other part of Introduction and Discussion 4.3), they focused on plot/stand scale mainly, not a catchment scale. SC1: 2. A brief paragraph should be added before Line 74 for an introduction of relevant studies that have been carried out in the Pearl River Basin. Without this, it is a bit out of blue to see the next paragraph suddenly mentioning something in this basin. AC: Thanks for the suggestion. A short paragraph has been added, please refer to Line 73-80 in the revised version, and the Study area section in 2.1 has been edited accordingly. SC1: 3. Regarding the data: I see a comparison between GLDAS precipitation and the ground truth data over a number of pixels given in Fig. 11. GRACE data from different processing centers are also compared. No comparisons/discussions are given for ETp and other variables. Can you find some studies in this basin or a basin with similar vegetation cover and climate that use GPP

from MODIS? If there is any, it'd provide more confidence in the results of this study. AC: Data uncertainty is a concern indeed, and some discussion has been given in section 4.1. Here we did not compare ETp, NDVI and GPP from variable sources, but just chose ETp from GLDAS, NDVI from GIMMS3g and GPP from MODIS, because during literature review, we found these three data are applied widely and commonly among studies, which would benefit comparisons between this study and others. Additionally, we provide two supplementary figures to show the comparisons of GPP based on multiple datasets. Justification is described in Line 130-139 and 261-275. SC1: 4. The current m/s is a complete story by overlooking the water-vegetation relationships in the entire basin in space and time. It is good to locate the hotspots of changes and interactions because these areas would usually be the 'focus' of land/water management and for risk control, etc. I recommend the authors to take a further step to investigate the reasons behind the changes and interactions right in these hotspot areas. AC: Thanks for the comment and suggestion. We are actually taking this hotspot out as an individual project in order to investigate in depth what drives the intensive changes of vegetation index and productivity in these areas. This study explains it from the perspective of water resources and climate dryness, and we will further explore the role of planting structure, agricultural management (including irrigation and fertilization), droughts, etc. Therefore, we decided not to extend further in this manuscript. SC1: 5. Paragraph ends with Line 279: This is a good argument that vegetation relies on water because of the lags of vegetation parameters after water input & storage dynamic change. However, there seems a lack of support to the opposite standing, i.e. vegetation growth does not result in excessive water reduction. So this part of discussion needs a further expansion. AC: From the phase shift between water (P & TWS) and vegetation growth (NDVI) at the monthly scale, we concluded that water limits vegetation growth in this region because the latter varies following the change in the former. The opposite possibility, i.e. vegetation water uptake leading to storage reduction cannot be detected at the investigated time scale but might be more evident at a shorter time scale like sub-daily in Kirchner et al., (2020) and Shen et al.,

(2015), who found decline in groundwater level/soil water content with increase in sap flow rates. Statement has been given in the relevant location in the text. Kirchner, J., Godsey, S., Osterhuber, R., McConnell, J. and Penna, D.: The pulse of a montane ecosystem: coupled daily cycles in solar flux, snowmelt, transpiration, groundwater, and streamflow at Sagehen and Independence Creeks, Sierra Nevada, USA, Hydrol. Earth Syst. Sci. Discuss., 1–46, doi:10.5194/hess-2020-77, 2020. Shen, Q., Gao, G., Fu, B. and Lü, Y.: Sap flow and water use sources of shelter-belt trees in an arid inland river basin of Northwest China, Ecohydrology, 8(8), 1446–1458, doi:10.1002/eco.1593, 2015. SC1: 6. Fig. 2-5, 7: the spatial distributions of these variables/trends are shown for all pixels. How would it be like if only the ones with $p<0.05$ are shown? AC: We meant to show the pixels with $p<0.05$ initially, but eventually decided to show all. In the revised manuscript, we have labelled the pixels with NDVI to show clearly the hotspot of change. The reduced number of pixels with $p<0.05$ can be a result of spatial resampling of data from a finer resolution to the current 0.5-degree resolution.

Please also note the supplement to this comment:
https://hess.copernicus.org/preprints/hess-2020-242/hess-2020-242-AC1-supplement.pdf

**Supplement:**

**Supplementary materials for hess-2020-242 '*Assessing the large-scale plant-water relations using remote sensing products in the humid subtropical Pearl River Basin in south China*'**

Monthly gross primary production (GPP) are obtained from MODIS (MOD17A2) (Running et al., 2004), VPM (Zhang et al., 2017) and PML (Zhang et al., 2019) where spatial differences exist (Fig. S1). The pattern between MODIS and PML is similar and differs from VPM where the latter shows much lower GPP anomalies spatially. Regardless of the differences in spatial distribution, GPP anomalies from the three sources agree well with correlation coefficient R>0.9 in most pixels. Temporally, the $R^2$ is 0.93, 0.94 and 0.96 between MODIS and VPM, MODIS and PML, and VPM and PML, respectively.

[Figure]

Figure S1. Comparison of monthly GPP anomaly from different sources based on different algorithms, i.e. MODIS, VPM and PML. R is the correlation coefficient between GPP from each two sources.

Fig. S2 shows the comparisons of monthly and annual GPP anomaly from MODIS, VPM and PML over the entire basin during the period of 04/2002-03/2015. It is observed that at the monthly scale GPP anomaly from MODIS is close to that from PML, whilst GPP anomaly from VPM is clearly lower than the other two, especially the median value. At the annual scale, the mean GPP anomaly is similar between

MODIS and PML and higher than that from VPM. Median value of MODIS is slightly higher than that of PML. Moreover, the data range of VPM is greater than that of MODIS and PML, which infers that VPM gives lower GPP values beyond growing seasons and higher values in growing seasons, as is shown in Fig. S1g.

[Figure]

Figure S2. Comparison of mean GPP based on MODIS, VPM and PML algorithms over the entire basin. Solid diamonds mark the mean GPP of each dataset. Unit for GPP is $gCm^{-2}$.

These comparisons of GPP from different sources demonstrate that the GPP values from MODIS and PML are comparable and VPM might underestimate GPP. However, without ground observations in the basin to validate these datasets, it is hard to conclude which dataset is the most accurate. Despite the accuracy issue, it should be similar when analyzing spatiotemporal relationships with hydroclimate variables using MODIS and PML data.

**References**

Running, S. W., Heinsch, F. A., Zhao, M., Reeves, M., Hashimoto, H. and Nemani, R. R.: A Continuous Satellite-Derived Measure of Global Terrestrial Primary Production., Bioscience, 54(6), 547–560, 2004.

Zhang, Y., Xiao, X., Wu, X., Zhou, S., Zhang, G., Qin, Y. and Dong, J.: A global moderate resolution dataset of gross primary production of vegetation for 2000-2016, Sci. data, 4, 170165, doi:10.1038/sdata.2017.165, 2017.

Zhang, Y., Kong, D., Gan, R., Chiew, F. H. S., McVicar, T. R., Zhang, Q. and Yang, Y.: Coupled estimation of 500 m and 8-day resolution global evapotranspiration and gross primary production in 2002–2017, Remote Sens. Environ., 222, 165–182, doi:10.1016/j.rse.2018.12.031, 2019.

---

## Author Comment (AC2) · 17 Sep 2020

RC1: The manuscript evaluates regional scale plant-water relations in the Pearl River Basin. The authors find a strong inter-annual correspondence between NDVI and GRACE derived TWS, suggesting water limitation in an area where rainfall is generally higher than the potential evapotranspiration. This is an interesting result, but the underlying mechanism remains unclear. The introduction touched on a few important topics such as water limitation and plant water use, but the scientific hypothesis/questions are not clearly defined. "Quantifying the plant-water relations at different temporal scales under different dryness conditions" is a good starting point,

but the specific questions to address need to be defined. The choice of vegetation data needs justification. NDVI is known to saturate in the forest ecosystem. MODIS GPP poorly represents soil moisture limitation on productivity which is directly relevant to the main theme of this study. There are many other vegetation metrics available that are not or less affected by these issues (e.g., SIF and EVI). LAI has also been used in a similar domain (Tong et al., 2018). I suggest the authors adopt these other datasets in the analysis. The strong inter-annual correspondence between NDVI and TWS is interesting, given how humid this area is. It would be of interest to see if this correspondence changes across different biomes (e.g. crops vs. forests) or regions with different levels of aridity, which may be done at mascon resolution. On the other hand, the monthly-scale correlation analysis needs clarification. Is the trend and seasonality removed from the monthly time series? The discussion session lacks a clear focus and sometimes reads like a literature review (e.g. Line 280-294). The discussion should be centered on clearly defined research questions and based directly on the results of this study. AC: We thank you for the recognition of our work and kindly pointing out the weaknesses for us to further improve the manuscript quality. We have stated clearly the specified research questions (reduced to 2 now) and discussed the possible underlying mechanisms (which is still limited based on the analysis) for the relationships from the perspective of energy & water availability in this environment compared to the dry environment. The mechanisms can be obtained with such comparisons but can hardly be verified using the applied data in this study. We have added a few words for the clarification in the Discussion. We were aware that applying different datasets (for both hydroclimate and vegetation) could lead to a possibly different result, therefore, we gave the reasons of our data choice in 4.1 Uncertainties in the datasets and results. Choice of vegetation data was based on literature review, that we found GIMMS NDVI3g is among the most popular datasets for analysis of vegetation phenology and its relationship with hydroclimate change, especially for studies in a relatively large river basin as it covers a moderately long time period (since 1980s) and has a spatial resolution of 1/12 degree. Using

GIMMS NDVI3g may allow the comparison of results in this study with many other studies in the region. Considering most of the forests consist of evergreen trees, and forest cover ( $\sim$ 65%) nearly remains constant from the early 21st century, the NDVI trend is highly likely induced primarily by other land cover types especially croplands ( $\sim$ 18%) and grassland ( $\sim$ 9%). For this reason, we think NDVI is fit for the purpose of this study. Using other vegetation indices like EVI and SIF may result in slightly different values of the trends but the overall changing direction (+/-) may be consistent. As to GPP data, we chose MOD17 products and the reasons are also given in session 4.1 Uncertainties in the datasets and results. This part of discussion has been extended with the assistance of inter-comparison of GPP from MODIS, VPM and PML given in the supplementary figure. Again, we thank you very much for the detailed comments which help improve our manuscript to a large degree. They have been carefully incorporated in the revised version. RC1: Detailed comments: Lines 72-73. This statement needs clarification. Is it to guestion if water limitation prevails in the humid ecosystems in the long term? AC: This sentence has been rephrased as 'While majority of such studies were carried out in semi-arid regions because of the urgent need to find an equilibrium between ecological restoration and available water resources in these water-limited areas, it is still largely unclear whether the restriction of water resources on vegetation growth also prevails in the humid or semi-humid areas with abundant rainfall.' RC1: Line 105. I think it is better to define the TWS anomaly using the entire analyzed period as a baseline (by removing the mean calculated over the entire period), unless there are specific reasons to believe that the 2004-2009 period better represents a "normal" condition. AC: GRACE satellite data are released by three processing centres as TWS anomaly which is the actual (ungiven) TWS value in each month minus the monthly mean from 2004 to 2009. To be consistent with the TWSA calculation norm, we also use the means of other variables in the same baseline period to obtain their anomalies. There is a good reason to question the representativeness of this period, however, it should be equally effective when analyzing their trends and relationships. RC1: Lines 129-132. The mean annual

СЗ

TWSA depends on the choice of the reference period. The trend analysis is a better way to illustrate wetting/drying information. Are all the trends significant in Fig 2d? AC: Indeed the mean annual TWSA and the annual trend would vary with the reference period. Here we show the values over 04/2002-03/2015 totaling 13 hydrologic years. Areas with p>0.05 in Figure 3d are distributed in the west edge where the annual trends are fairly small. Because it takes a small proportion of the entire basin, we did not take these pixels out. RC1: Fig 2e. Please clarify how the basin average and the associated errors (measurement and leakage) are calculated. This should be included in the Method session. AC: Thanks for the suggestion. We have added information in the 1st paragraph of subsession 2.2. Data sources and pre-processing as suggested. Please refer to Line 119-122. RC1: Line 145. What is the trend in space? Note that here the trend in time does not have an error bar. AC: We have rephrased this sentence as '... with an overall positive trend spatially ( $0.004\pm0.012$ ) and temporally  $(0.007\pm0.028), \ldots$  Spatially, we calculated the linear trend for each valid pixel, and then the average of all with the standard deviation. RC1: Figs 3-5. Please change the color scheme to improve the readability of the figures. For example, a sequential colormap is ideal for the aridity index. For the anomaly and trends, it is better to use a diverging colormap with a symmetric scale. AC: Thank you for the suggestion. We have changed the color scheme accordingly to improve the figure readability. Please refer to the updated figures. RC1: Line 153. Please label the significant trends in the map. AC: The NDVI trend map has been reproduced as below with pixels of p<0.05 marked with crosses. RC1: Lines 159-161. This reads like discussion, not actual results. AC: We have carefully checked the results and moved the discussion-like contents to Discussion. RC1: Lines 169-170. Needs other proxies for plant productivity to confirm this. MODIS GPP directly accounts for the limitation from VPD but not from soil moisture supply. AC: Thank you for the suggestion. Indeed, MODIS GPP does not account for moisture constraint but rather atmospheric controls. In our study area, rainfall and water storage is high in the growing seasons and slightly lower in the nongrowing seasons. In this case, the moisture restriction on GPP might be small.

In addition, we compare GPP from three sources (MODIS, VPM and PML) in the supplementary figures, and with that support, we still use MODIS GPP in the analysis. Session 2.2 regarding data sources and session 4.1 regarding data uncertainty are extended to incorporate this content (Line 130-139 and 261-275). RC1: Fig 7. Please either label the areas with significant correlations or mask the insignificant ones. Trends can inflate the correlation results. Have you de-trended the time series? AC: Linear trends are removed from the correlation analysis. This has been added in the Method session (Line 148-149). Thanks. RC1: Line 182. It is unclear how the monthly scale regression is calculated. Note that to quantify water limitation, the seasonality should be removed from the monthly time series. AC: Information has been added in the Method session (Line 148-149). Data were detrended before calculating the correlation coefficients. RC1: Lines 189-190. It is unclear what this means. How are the water restriction and water consumption quantified and compared? In fact, quantifying the amount and timing of plant water consumption (e.g. ET in wet and dry years) might be helpful to understand why there is an apparent water restriction in such a humid area. AC: This sentense has been rephrased (Line 221-223). We meant to show that the influence of water availability on vegetation development may be stronger than the impact of vegetation growth on water resources (mostly water storage), because decline of TWSA occurred prior to reduction of NDVI. RC1: Line 196. How is the span of the growing season defined in this area? AC: Growing season months have been given in Method session (Line 138). Because it can vary from year to year for each type of vegetation cover, we use the conventional definition in this study, i.e. from April to October. Precise quantification of growing season length can be done with NDVI/LAI time series but won't be necessary for this study. RC1: Lines 212-220. This should go to the Data and Method session. AC: We have moved this part to Data and Methods right after introduction of GLDAS precipitation. RC1: Line 230. The uncertainty of the trend needs to be evaluated. AC: Uncertainty has been added. RC1: Lines 232-241. This should go to the Data and Method session. The authors present examples where MODIS GPP shows consistency with other vegetation

data, but in thisstudy, the analysis based on the two datasets (MODIS GPP and NDVI) shows different plant-water relations. It is unclear if the difference is physical (e.g. due to the different responses of vegetation state and vegetation productivity) or caused by data accuracy issues. In this case, other vegetation metrics are needed to justify the results. AC: We have added a few sentences in Data and Method about GPP data, and also extended a bit in this part of Discussion (Line 130-139 and 261-275). RC1: Line 257. Note that this is an active area for ecological restoration, including the Grain to Green project (Tong et al., 2018). Reference: Tong, X., Brandt, M., Yue, Y., Horion, S., Wang, K., Keersmaecker, W. De, : : : Fensholt, R. (2018). 1. Nature Sustainability, 1(1), 44-50. https://doi.org/10.1038/s41893-017-0004-x AC: Noted and incorporated into disucssoin (Line 289-292). In their Fig. 3, Tong et al. mapped the convervation efforts in their study area most of which show low-moderate levels. They also show increasing trends of LAI in the region where croplands dominate (lower right part of their study area, with low-moderate conversation level). This indirectly supports our finding that the vegetation growth in this cultivated area has been enhanced. RC1: Lines 272-275. This point seems important but is not fully developed. Are there results in this study showing enhanced or perhaps near-normal productivity under drier than normal condition? AC: We have rephrased this part (Line 311-316).The comparisons of NDVI, GPP in dry and wet years in Fig. 10b-c and the relationships between them and P/TWS in Fig. 11 support this speculation. NDVI in dry and wet years was largely different, while GPP in dry years was only slightly lower than that in wet years, showing a less sensitive response of GPP to drying than that of NDVI.

Please also note the supplement to this comment: https://hess.copernicus.org/preprints/hess-2020-242/hess-2020-242-AC2supplement.pdf

**Supplement:**

**Supplementary materials for hess-2020-242 '*Assessing the large-scale plant-water relations using remote sensing products in the humid subtropical Pearl River Basin in south China*'**

Monthly gross primary production (GPP) are obtained from MODIS (MOD17A2) (Running et al., 2004), VPM (Zhang et al., 2017) and PML (Zhang et al., 2019) where spatial differences exist (Fig. S1). The pattern between MODIS and PML is similar and differs from VPM where the latter shows much lower GPP anomalies spatially. Regardless of the differences in spatial distribution, GPP anomalies from the three sources agree well with correlation coefficient R>0.9 in most pixels. Temporally, the $R^2$ is 0.93, 0.94 and 0.96 between MODIS and VPM, MODIS and PML, and VPM and PML, respectively.

[Figure]

Figure S1. Comparison of monthly GPP anomaly from different sources based on different algorithms, i.e. MODIS, VPM and PML. R is the correlation coefficient between GPP from each two sources.

Fig. S2 shows the comparisons of monthly and annual GPP anomaly from MODIS, VPM and PML over the entire basin during the period of 04/2002-03/2015. It is observed that at the monthly scale GPP anomaly from MODIS is close to that from PML, whilst GPP anomaly from VPM is clearly lower than the other two, especially the median value. At the annual scale, the mean GPP anomaly is similar between

MODIS and PML and higher than that from VPM. Median value of MODIS is slightly higher than that of PML. Moreover, the data range of VPM is greater than that of MODIS and PML, which infers that VPM gives lower GPP values beyond growing seasons and higher values in growing seasons, as is shown in Fig. S1g.

[Figure]

Figure S2. Comparison of mean GPP based on MODIS, VPM and PML algorithms over the entire basin. Solid diamonds mark the mean GPP of each dataset. Unit for GPP is $gCm^{-2}$.

These comparisons of GPP from different sources demonstrate that the GPP values from MODIS and PML are comparable and VPM might underestimate GPP. However, without ground observations in the basin to validate these datasets, it is hard to conclude which dataset is the most accurate. Despite the accuracy issue, it should be similar when analyzing spatiotemporal relationships with hydroclimate variables using MODIS and PML data.

**References**

Running, S. W., Heinsch, F. A., Zhao, M., Reeves, M., Hashimoto, H. and Nemani, R. R.: A Continuous Satellite-Derived Measure of Global Terrestrial Primary Production., Bioscience, 54(6), 547–560, 2004.

Zhang, Y., Xiao, X., Wu, X., Zhou, S., Zhang, G., Qin, Y. and Dong, J.: A global moderate resolution dataset of gross primary production of vegetation for 2000-2016, Sci. data, 4, 170165, doi:10.1038/sdata.2017.165, 2017.

Zhang, Y., Kong, D., Gan, R., Chiew, F. H. S., McVicar, T. R., Zhang, Q. and Yang, Y.: Coupled estimation of 500 m and 8-day resolution global evapotranspiration and gross primary production in 2002–2017, Remote Sens. Environ., 222, 165–182, doi:10.1016/j.rse.2018.12.031, 2019.

---

## Author Comment (AC3) · 17 Sep 2020

RC2: General comments: 1. The Pearl River Basin is in relatively humid region. Beside water, other factors may also influence the vegetation growth. It is suggested to show the landcover change in the studied period and analyze the relationship between vegetation growth and temperature or egergy to identify the vegetation-water relation more clearly. AC: Thank you for the suggestion. It is a good one and in fact we thought about this analysis, because it is realized that the controling factors of vegetation growth can be divided into two groups – the demand (including radiation, vapor pressure deficit, and temperature, etc) and the supply groups (soil moisture, groundwater,

and water storage, etc). The supply factor was represented by precipitation and total water storage here, and the demand effect was integrated in potential evaporation and embedded in the aridity index. In this sense, we discussed both the hydroclimate and water impacts on vegetation. We made the argument more clearly at the end of Methods section (Line 156-160). RC2: 2. Lag effect between vegetation growth and water availability are analyzed at monthly scale. In my opinion, it is necessary to show how P, TWS, NDVI and GPP for the 12 months in a year for better discussion about the lag effect. AC: We agree that a monthly climatological mean of these variables would help much with the lag effect analysis. We also gave this calculation and analysis in Figure 10, and results in Figure 11 were based on the climatological means which we failed to mention in the relevant text. Please refer to Line 236 in the revised version. RC2: 3. I understand when using remote sensing products, uncertainty issue is always a concern need to be addressed. However, this is not the scientific target of this paper. To keep the readers' attention to the key scientific question trying to answer, it is suggested to remove the "uncertainties in the datasets and results" section and describe how you quantify the uncertainty of remote sensing data in the Methodology section. AC: Thanks for the suggestion. Indeed, when using remote sensing for hydrologic studies, the data uncertainty/accuracy is often concerned. Considering that the other anonymous reviewer also cared about this, we still keep this subsection in the revised manuscript and expand it for a detailed justification. A few supplementary figures are also provided for comparisons of different remote sensing products. Relevant text in Data and methods and Discussion has been improved (Line 130-139 and 261-275). RC2: Specific comments: Line 77: Please give more information about the importance of Pearl River Basin and it's connection with research progress described in the previous paragraph. AC: This issue is also suggested by another reviewer. A short paragraph has been added, please refer to Line 73-80 in the revised manuscript, and the Study area section in 2.1 has been edited accordingly as well. Because more information about previous relevant studies were to be mentioned in the Study area and Discussion sections, we only talked about the basin in general

here to avoid the abrupt shift in to the Pearl River Basin in this part of Introduction. RC2: Line 120: It is suggested to decide the assumption being made behind the lag effect analysis AC: We have changed the sentence 'Furthermore, a lag effect analysis …' to 'Furthermore, to investigate the causal role of vegetation growth to water storage changes (or the vice versa), we carried out lag effect analysis between vegetation parameters and hydroclimate variables'. RC2: Line 195: The basin is in subtropical region. So please confirm whether October to March is non-growing seasons. AC: Growing season months have been given in Method session (Line 138). Because it can vary from year to year for each type of vegetation cover, we use the conventional definition in this study, i.e. from April to October. Precise quantification of growing season length can be done with NDVI/LAI time series but won't be necessary for this study. RC2: Line 252-253: A landcover change analysis for the study period may make the explanation here more persuasive. AC: Please refer to the response to General comment point 1, and response to the 4th comment by Prof. Zhang. In addition, we think the possible changes in planting structure would also alter the trend of greenness and productivity in these agricultural areas. RC2: Line 254: I'm a little bit confused about "water storage increase in this hotspot region has resulted in the intensification of agricultural activities". More explanation is needed. AC: We have rephrased this sentence as 'The changes of TWS, NDVI and GPP jointly imply that the water storage increase in this hotspot region, which may be induced by increased precipitation, corresponds to the intensification of agricultural activities and boosted the food production since the early 2000s.' A study by Tong et al., (2018) was used to partly support our finding here. RC2: Figure 9: It is hard to read as many elements are overlapped together. Please find a clearer way to describe the information contained in this figure. AC: We have separated Figure 10a in 2 subplots, and adjusted the colors and transparency of the bands to show them as clearly as possible.

Please also note the supplement to this comment:
https://hess.copernicus.org/preprints/hess-2020-242/hess-2020-242-AC3-

supplement.pdf

---

## Author Response (AR1)

We are grateful for the detailed and careful review of our work by the three referees and the Editor Prof. . The following lists the responses to these comments and suggestions. The resubmitted manuscript has been corrected and improved accordingly.

**Comments by Prof. Qiang Zhang**

**SC1:** Water is one of the critical resources to sustain the rapid socioeconomic development. Vegetation cover is high and most of them is evergreen subtropical species, which means there could be substantial water consumption by plants throughout the year given the favourable climate. Water and ecological management faces increasing challenges because of the rapid population growth, high urbanization and industrialization, etc. Therefore, this is a timely study to investigate the intensive changes and interactions between vegetation and water. The most interesting part of this study, to me, is the identification of the 'hotspot' of changes and determination of water-limiting-vegetation even in this rainfall-abundant region. The m/s is well structured, the wordings are fine, and method/analysis are appropriate. In my view, this study is meaningful and worth publishing. Having said these, I do still have a few concerns listed below for the authors to address:

Thank you for the encouraging comments. The concerns are addressed below.

**SC1:** 1. I find the summary of the water-plant relationships in Line 61-69 is quite interesting. Vegetation consumes water and causes reduction in water resources on the one hand, and water availability will restrict vegetation establishment and growth on the other. Indeed, plant-water relations are examined mostly in arid and semi-arid regions for the purpose of water and ecological conservation. Are there such studies in humid and semi-humid regions investigating the controlling factors – energy vs. water - of vegetation growth? It is important and is the authors responsibility to ensure a thorough literature review on this subject.

AC: The possibility of different vegetation-water relationships under contrast climate conditions is the motivation of this study.

Most such studies focus on the drylands because of the likely more severe water scarcity and ecological problems, while the plants and water/energy relationships are left less clear in the subtropical wet/humid areas where precipitation and radiation are both abundant. Although there have been studies in humid areas investigating environmental controls on plant water use such as those in the last paragraph of Introduction and Discussion 4.3, they focused on plot/stand scale mainly, or a national scale, and rarely discussed the relationships under contrasting dryness conditions.

**SC1:** 2. A brief paragraph should be added before Line 74 for an introduction of relevant studies that have been carried out in the Pearl River Basin. Without this, it is a bit out of blue to see the next paragraph suddenly mentioning something in this basin.

AC: Thanks for the suggestion. A short paragraph has been added, please refer to Line

78-89 in the revised version, and the Study area section 2.1 has been edited accordingly.

**SC1:** 3. Regarding the data: I see a comparison between GLDAS precipitation and the ground truth data over a number of pixels given in Fig. 11. GRACE data from different processing centers are also compared. No comparisons/discussions are given for ETp and other variables. Can you find some studies in this basin or a basin with similar vegetation cover and climate that use GPP from MODIS? If there is any, it'd provide more confidence in the results of this study.

AC: Data uncertainty is always a big concern especially when remote sensing and modelling results are involved. In this submission, we have obtained ETp and GPP from more common sources and provided more comparisons to discuss the data uncertainty. The comparisons are given in a supplementary document and referenced in the main text. Please refer to Section 2.2 *Data sources and pre-processing*, and section 4.1 *Uncertainties in the datasets and results*.

**SC1:** 4. The current m/s is a complete story by overlooking the water-vegetation relationships in the entire basin in space and time. It is good to locate the hotspots of changes and interactions because these areas would usually be the 'focus' of land/water management and for risk control, etc. I recommend the authors to take a further step to investigate the reasons behind the changes and interactions right in these hotspot areas.

AC: Thanks for the comment and suggestion. The main purpose of this study is to examine the relationships between vegetation parameters and hydroclimates, especially under contrasting atmospheric dryness conditions. Through the analysis, we found the areas of croplands where vegetation parameters and hydroclimates changed greater than other areas, and presumed that the relationship in these areas is possibly related to agricultural activities like irrigation and planting structure change, etc. We screen this hotspot area out in another work to investigate in depth what drives the intensive changes of vegetation index and productivity in these areas. For that we are still collecting agricultural data including planting structure, crop yield, cropland area, irrigation and fertilization areas, etc. They are not incorporated into this study.

**SC1:** 5. Paragraph ends with Line 279: This is a good argument that vegetation relies on water because of the lags of vegetation parameters after water input & storage dynamic change. However, there seems a lack of support to the opposite standing, i.e. vegetation growth does not result in excessive water reduction. So this part of discussion needs a further expansion.

AC: From the phase shift between water (P & TWS) and vegetation growth at the monthly scale, we concluded that water limits vegetation growth in this region because the latter varies following the change in the former. The opposite possibility, i.e. vegetation water uptake leading to storage reduction cannot be detected at the investigated time scale but might be more evident at a shorter time scale like sub-daily in Kirchner et al., (2020) and Shen et al., (2015), who found decline in groundwater level/soil water content with increase in sap flow rates. Statement has been given in the

relevant location in the text (Line 354-357).

Kirchner, J., Godsey, S., Osterhuber, R., McConnell, J. and Penna, D.: The pulse of a montane ecosystem: coupled daily cycles in solar flux, snowmelt, transpiration, groundwater, and streamflow at Sagehen and Independence Creeks, Sierra Nevada, USA, Hydrol. Earth Syst. Sci. Discuss., 1–46, doi:10.5194/hess-2020-77, 2020.

Shen, Q., Gao, G., Fu, B. and Lü, Y.: Sap flow and water use sources of shelter-belt trees in an arid inland river basin of Northwest China, Ecohydrology, 8(8), 1446–1458, doi:10.1002/eco.1593, 2015.

**SC1:** 6. Fig. 2-5, 7: the spatial distributions of these variables/trends are shown for all pixels. How would it be like if only the ones with p<0.05 are shown?

AC: Trends of some variables are not statistically significant. We have marked the pixels with significant trends for NDVI and GPP in Fig. 4-5. The correlation coefficient with p<0.05 is also marked in Fig. 7.

**Anonymous Referee #1:**

**RC1**: The manuscript evaluates regional scale plant-water relations in the Pearl River Basin. The authors find a strong inter-annual correspondence between NDVI and GRACE derived TWS, suggesting water limitation in an area where rainfall is generally higher than the potential evapotranspiration. This is an interesting result, but the underlying mechanism remains unclear. The introduction touched on a few important topics such as water limitation and plant water use, but the scientific hypothesis/questions are not clearly defined. "Quantifying the plant-water relations at different temporal scales under different dryness conditions" is a good starting point, but the specific questions to address need to be defined.

The choice of vegetation data needs justification. NDVI is known to saturate in the forest ecosystem. MODIS GPP poorly represents soil moisture limitation on productivity which is directly relevant to the main theme of this study. There are many other vegetation metrics available that are not or less affected by these issues (e.g., SIF and EVI). LAI has also been used in a similar domain (Tong et al., 2018). I suggest the authors adopt these other datasets in the analysis.

The strong inter-annual correspondence between NDVI and TWS is interesting, given how humid this area is. It would be of interest to see if this correspondence changes across different biomes (e.g. crops vs. forests) or regions with different levels of aridity, which may be done at mascon resolution. On the other hand, the monthly-scale correlation analysis needs clarification. Is the trend and seasonality removed from the monthly time series?

The discussion session lacks a clear focus and sometimes reads like a literature review (e.g. Line 280-294). The discussion should be centered on clearly defined research

questions and based directly on the results of this study.

AC: We thank you for kindly pointing out the weaknesses for us to further improve the manuscript quality.

In the Introduction, we reviewed studies of plant-water relationships using both field observations and remote sensing across different spatial scales and summarized some general findings of such studies. Further, we stressed that these findings are mostly based on studies in the arid and semi-arid regions, while studies in radiation-sufficient humid and semi-humid regions are still limited. We have restated the specified research objectives and discussed the possible underlying mechanisms (which is still limited based on the analysis) for the relationships from the perspective of energy & water availability in this environment compared to the dry environment. The mechanisms can be obtained with such comparisons but can hardly be verified using the applied data in this study.

We were aware that applying different datasets (for both hydroclimate and vegetation) could lead to a possibly different result, therefore, we gave the reasons of our data choice in *2.2 Data sources and pre-processing* and *4.1 Uncertainties in the datasets and results*. Choice of vegetation data was based on literature review, that we found GIMMS NDVI3g is among the most popular datasets for analysis of vegetation phenology and its relationship with hydroclimate change, especially for studies in a relatively large river basin as it covers a moderately long time period (since 1980s). Using GIMMS NDVI3g may allow the comparison of this study with many other studies in the region. Considering most of the forests consist of evergreen trees, and forest cover (~65%) nearly remains constant from the early 21$^{st}$ century, the NDVI trend is highly likely induced primarily by other land cover types especially croplands (~18%) and grassland (~9%). From these points of view, we think NDVI is fit for the purpose of this study. Using other vegetation indices like EVI and SIF may result in slightly different values of the trends but the overall changing direction (+/-) may be consistent. As to GPP data, we have added data from VPM and PML in addition to MOD17 products and the comparisons among them are given in section 4.1 and the supplementary document. We found the overall results (spatial distribution, trends and relationships with hydroclimate data) changed little compared to the results based solely on MODIS GPP.

We have also improved the discussion with an emphasis on data uncertainty, hotspots for changes and possible reasons, as well as the interactions in dry and wet conditions.

**RC1:** Detailed comments:

Lines 72-73. This statement needs clarification. Is it to question if water limitation prevails in the humid ecosystems in the long term?

AC: This sentence has been rephrased as '*While majority of such studies were carried out in semi-arid regions because of the urgent need to find an equilibrium threshold*

*between ecological restoration and available water resources in these water-limited areas, it is still largely unclear whether the restriction of water resources or available energy on vegetation growth prevails in the humid or semi-humid areas with both abundant rainfall and radiation*'

**RC1:** Line 105. I think it is better to define the TWS anomaly using the entire analyzed period as a baseline (by removing the mean calculated over the entire period), unless there are specific reasons to believe that the 2004-2009 period better represents a "normal" condition.

AC: We did not define the baseline period for GRACE data. Actually, GRACE satellite data are released by three processing centres as TWS anomaly, which is the actual (ungiven) TWS value in each month minus the monthly mean from the period of 2004 to 2009. There is a good reason to question the representativeness of this period as 'normal' condition, but with all data relative to the same baseline period, we believe the results will not be affected.

**RC1:** Lines 129-132. The mean annual TWSA depends on the choice of the reference period. The trend analysis is a better way to illustrate wetting/drying information. Are all the trends significant in Fig 2d?

AC: Refer to the above respones. In addition, the linear trend will not change after subtracting a value from a data series, even if the minuend differs. So the trend analysis is not affected by the baseline period.

The spatial TWSA trends are mostly insignificant, just like its temporal trends. The trend will change with the study period though, for example, if we focus on the period of 04/2003-03/2015, then the linear trend will be 6.99 mm yr$^{-1}$, with a $p$ value of 0.006. This is mentioned in the Discussion 4.1 starting with Line 293.

**RC1:** Fig 2e. Please clarify how the basin average and the associated errors (measurement and leakage) are calculated. This should be included in the Method session.

AC: Thanks for the suggestion. We have added information in the 1$^{st}$ paragraph of subsession *2.2 Data sources and pre-processing* and second paragraph in *2.3 Data analysis* as suggested.

**RC1:** Line 145. What is the trend in space? Note that here the trend in time does not have an error bar.

AC: We have rephrased this sentence as '… with an overall positive trend spatially (0.002±0.009) and temporally (0.005±0.025), …'.

**RC1:** Figs 3-5. Please change the color scheme to improve the readability of the figures. For example, a sequential colormap is ideal for the aridity index. For the anomaly and trends, it is better to use a diverging colormap with a symmetric scale.

AC: Thank you for the suggestion. We have changed the color scheme accordingly to improve the figure readability. Please refer to the updated figures.

**RC1:** Line 153. Please label the significant trends in the map.

AC: The NDVI, GPP trend maps have been reproduced with pixels of $p<0.05$ marked with crosses. So has the figure for correlation coefficients of annual data.

**RC1:** Lines 159-161. This reads like discussion, not actual results.

AC: We have carefully checked the results and moved the discussion-like contents to Discussion.

**RC1:** Lines 169-170. Needs other proxies for plant productivity to confirm this. MODIS GPP directly accounts for the limitation from VPD but not from soil moisture supply.

AC: Thank you for the suggestion. Indeed, MODIS GPP alongside many other GPP products does not account for moisture constraint but rather atmospheric controls including temperature, VPD and radiation. In our study area, rainfall and water storage is high in the growing seasons (conventinally defined as April to October) and slightly lower in the nongrowing seasons. In this case, the moisture restriction on GPP might be small. In addition, we compare GPP from three sources (MODIS, VPM and PML) in the supplementary figures, and we used the mean GPP values of the three products in the this submission. Session 2.2 regarding data sources and session 4.1 regarding data uncertainty are extended to incorporate this content.

**RC1:** Fig 7. Please either label the areas with significant correlations or mask the insignificant ones. Trends can inflate the correlation results. Have you de-trended the time series?

AC: Linear trends are removed before the correlation analysis. This information has been added in the Method session (Line 159). Thanks. Fig. 7 has been updated with crosses marking the significant correlation coefficients.

**RC1:** Line 182. It is unclear how the monthly scale regression is calculated. Note that to quantify water limitation, the seasonality should be removed from the monthly time series.

AC: Information has been added in the Method session (Line 159). Data were detrended before calculating the correlation coefficients.

**RC1:** Lines 189-190. It is unclear what this means. How are the water restriction and water consumption quantified and compared? In fact, quantifying the amount and timing of plant water consumption (e.g. ET in wet and dry years) might be helpful to understand why there is an apparent water restriction in such a humid area.

AC: This sentense has been rephrased (Line 233-235). We assume that vegetation growth is constrained by water resources if dynamics of NDVI/GPP falls behind dynamics of P/TWSA (Line 164-167). The degree of constraints of dryness and water on vegetation is implied by the corelation coefficient in Fig. 6-7 and Fig. 10.

**RC1:** Line 196. How is the span of the growing season defined in this area?

AC: Growing season months have been given at their first appearance in section 3.3 (Line 241 & 246). Because it can vary from year to year for each type of vegetation cover, we use the conventional definition in this study, i.e. from April to October. Precise quantification of growing season length can be done with vegetation index time series but won't be necessary for this study.

**RC1:** Lines 212-220. This should go to the Data and Method session.

AC: We have provided comparisons of P, ETp, GPP from multiple sources in the section 2.2, and extened the discussion of data uncertainty in section 4.1.

**RC1:** Line 230. The uncertainty of the trend needs to be evaluated.

AC: The uncertainty of the temporal and spatial trend analysis throughout the text and figures has been defined. Please refer to the updated submission.

**RC1:** Lines 232-241. This should go to the Data and Method session. The authors present examples where MODIS GPP shows consistency with other vegetation data, but in thisstudy, the analysis based on the two datasets (MODIS GPP and NDVI) shows different plant-water relations. It is unclear if the difference is physical (e.g. due to the different responses of vegetation state and vegetation productivity) or caused by data accuracy issues. In this case, other vegetation metrics are needed to justify the results.

**AC:** In this submission, we used multiple datasets to reevaluate the relationships between vegetation and hydroclimate, and found that using the ensemble means of multiple datasets did not lead to significant difference in the results compared to the last submission. In addition, we noticed that GPP algorithms for MODIS, VPM and PML are all formulated with atmospheric variabels like temperature, VPD and radiation. It is found there exists time lags between these atmospheric conditions and NDVI, therefore, the NDVI and GPP should not synchronized in temporal dynamics, which would result in different response characteristics. This has been added in Discussion 4.3 (Line 334-339).

**RC1:** Line 257. Note that this is an active area for ecological restoration, including the Grain to Green project (Tong et al., 2018). Reference: Tong, X., Brandt, M., Yue, Y., Horion, S., Wang, K., Keersmaecker, W. De, : : : Fensholt, R. (2018). 1. Nature Sustainability, 1(1), 44–50. https://doi.org/10.1038/s41893-017-0004-x

**AC:** Noted and incorporated into disucssoin (Line 314-316). In their Fig. 3, Tong et al. mapped the convervation efforts in their study area most of which show low-moderate

levels. They also show increasing trends of LAI in the region where croplands dominate (lower right part of their study area, with low-moderate conversation level). This indirectly supports our finding that the vegetation growth in this cultivated area has been enhanced.

**RC1:** Lines 272-275. This point seems important but is not fully developed. Are there results in this study showing enhanced or perhaps near-normal productivity under drier than normal condition?

**AC:** We have rephrased this part (Line 340-343).The possible underlying mechanisms for higher GPP in dry conditions than wet conditions are also given in the follow-on test (Line 345-348).

**Anonymous Referee #2:**

**RC2:** General comments: 1. The Pearl River Basin is in relatively humid region. Beside water, other factors may also influence the vegetation growth. It is suggested to show the landcover change in the studied period and analyze the relationship between vegetation growth and temperature or egergy to identify the vegetation-water relation more clearly.

**AC:** Thank you for the suggestion. It is a good one and in fact we thought about this analysis, because it is realized that the controling factors of vegetation growth can be divided into two groups – the demand (including radiation, vapor pressure deficit, and temperature, etc) and the supply groups (soil moisture, groundwater, and water storage, etc).

The supply group factor was represented by precipitation and total water storage here, and the demand effect was integrated in potential evaporation and embedded in the aridity index. In this sense, we discussed both the hydroclimate and water impacts on vegetation. We made the argument more clearly at the end of Methods section (Line 173-176).

**RC2:** 2. Lag effect between vegetation growth and water availability are analyzed at monthly scale. In my opinion, it is necessary to show how P, TWS, NDVI and GPP for the 12 months in a year for better discussion about the lag effect.

**AC:** We agree that a climatological monthly mean of these variables would help much with the lag effect analysis. We also gave this calculation and analysis in Fig. 9, and results in Fig. 10 were based on the climatological means which we failed to mention in the relevant text. Please refer to Line 248-249 in the revised version.

**RC2:** 3. I understand when using remote sensing products, uncertainty issue is always a concern need to be addressed. However, this is not the scientific target of this paper. To keep the readers' attention to the key scientific question trying to answer, it is suggested to remove the "uncertainties in the datasets and results" section and describe

how you quantify the uncertainty of remote sensing data in the Methodology section.

**AC:** Thanks for the suggestion. Indeed, when using remote sensing for hydrologic studies, the data uncertainty/accuracy is often concerned. Considering that the other reviewer also mentioned this, we still kept this subsection in the revised manuscript and expand it for a detailed justification. A few supplementary figures are also provided for comparisons of different remote sensing products. Relevant text in Data and methods and Discussion has been improved.

**RC2:** Specific comments: Line 77: Please give more information about the importance of Pearl River Basin and it's connection with research progress described in the previous paragraph.

**AC:** This issue is also suggested by another reviewer. A short paragraph has been added, please refer to Line 78-89 in the revised manuscript, and the Study area section in 2.1 has been edited accordingly as well.

**RC2:** Line 120: It is suggested to decide the assumption being made behind the lag effect analysis

**AC:** We have changed the sentence '*Furthermore, a lag effect analysis ...*' to '*Furthermore, to investigate the causal role of vegetation growth to water availability changes (or vice versa), we carried out lag effect analysis between vegetation parameters and hydroclimate variables. That is, we assume that vegetation growth is constrained by water resources if dynamics of NDVI/GPP falls behind dynamics of P/TWSA.*' (Line 160-163)

**RC2:** Line 195: The basin is in subtropical region. So please confirm whether October to March is non-growing seasons.

**AC:** Growing season months have been given at its first appearance (Line 241 & 246). Because it can vary from year to year for each type of vegetation cover, we use the conventional definition in this study, i.e. from April to October. Precise quantification of growing season length can be done with vegetation index time series but won't be necessary for this study.

**RC2:** Line 252-253: A landcover change analysis for the study period may make the explanation here more persuasive.

**AC:** Please refer to the response to General comment point 1, and response to the 4th comment by Prof. Zhang. In addition, we think the possible changes in planting structure would also alter the trend of greenness and productivity in these agricultural areas (added discussion in Line 345-348).

**RC2:** Line 254: I'm a little bit confused about "water storage increase in this hotspot region has resulted in the intensification of agricultural activities". More explanation is needed.

**AC:** We have rephrased this sentence as '*The changes of TWS, NDVI and GPP jointly imply that the water storage increase in this hotspot region, which was likely induced by increased precipitation, coincides with the intensification of agricultural activities and boosted the food production since the early 2000s..*' A study by Tong et al., (2018) was used to partly support our finding here.

**RC2:** Figure 9: It is hard to read as many elements are overlapped together. Please find a clearer way to describe the information contained in this figure.

**AC:** We have separated Fig. 9a in 2 subplots, and adjusted the colors and transparency of the bands to show them as clearly as possible. Please refer to the revised figure.

[revised manuscript text omitted]
. Monthly TWSA is the result of subtracting the average TWS over the period of 01/2004-12/2009 from each monthly TWS value. In addition, GRACE$_{JPL}$ data uncertainties are given by these processing centres as the measurement and leakage errors for GRACE$_{JPL}$ (Swenson and Wahr, 2006; Wiese et al., 2016). In this study, when showing the basin-average monthly/annual TWSA dynamics, we used the standard deviation to define the uncertainty range for the entire basin.

140    Precipitation (P) and potential evapotranspiration (ETp) data were obtained from Global Land Data Assimilation System (GLDAS) ; (Rodell et al., 2004) and the national standard meteorological stations distributed across the basin from the China Meteorological Administration (CMA). Comparison of P from GLDAS and stations is given in the supplementary document Fig. S1, which shows that aridity index (AI) was then calculated as the ratio of ETp to P to represent the dryness condition. GLDAS uses meteorological forcing data merged from multiple sources including ground and satellite observations, and

145    GLDAS precipitation proves to be highly consistent with observations in China . Here we also compared the GLDAS P with the measured P in the pixels where stations are available (Fig. 11). Overall, P from GLDAS agreed well with observations with $R^2$ ranging from 0.69 to 0.89 (±0.05) spatially, while on average the monthly P from GLDAS slightly underestimated observations by ~10% over all valid pixels ($R^2$=0.98). The comparison provides some confidence in applying the gridded GLDAS productsP for long-term and spatial hydrological trend analysis in this basin, though discrepancies exist in the

150    absolute values. Potential evapotranspiration (ETp) was obtained from the GLDAS, and MODIS and PML, and comparisons among them are given in Fig. S2-3, which show that xxxboth products show ETp has been increasing over the 13 years, although GLDAS gave generally higher ETp than MODIS. GLDAS shows that ETp increase was largest over the croplands in the middle-south of the basin. Spatially, the correlation coefficient between these two ETp datasets ranges from 0.26 to 0.87 at the monthly scale and -0.11 to 0.76 at the annual scale. Temporally, the average ETp from GLDAS is 1579±1023.7

155    and 1504±11.54 mm yr$^{-1}$, and the coefficient of determination ($R^2$) between ETp from the two sources is 0.58 and 0.51 at the monthly and annual scale, respectively.

Total water storage (TWS) change is inferred by the mass change detected by GRACE satellites . GRACE data can be accessed from the Jet Propulsion Laboratory (JPL), the Center for Space Research (CSR), and the German Research Centre for Geosciences. Previous studies have shown that the ensemble mean of different products is effective in reducing the noise in the gravity field solutions . Here we used total water storage anomaly (TWSA) data from the JPL and CSR with 'mascons'

solution (release 6) at a resolution of 0.5° and monthly. Cubic spline interpolation was applied to estimate the missing monthly data for the GRACE_JPL and GRACE_CSR products during 04/2002-03/2015 that cover 13 hydrological years. To reduce the effect of errors embedded in each individual product, we calculated the average ETp from the  two sources for later analysis. Aridity index (AI) was then calculated as the ratio of ETp to P to represent the atmospheric dryness condition.

Vegetation data  include Normalized Difference Vegetation Index (NDVI) and Gross Primary Production (GPP) representing surface greenness and productivity, respectively. NDVI was obtained from the GIMMS project at a 15-day and 1/12° resolution  and resampled to 0.5° using the nearest neighbour method, and then averaged to monthly . GIMMS NDVI is among the most popular vegetation index datasets for analysis of vegetation phenology and its relationship with hydroclimate change (Cong et al., 2013; Jeong et al., 2011), especially for studies in a relatively large river basin as it covers a moderately long time period (since 1980s). Monthly GPP was obtained from the Numerical Terradynamic Simulation Group in the University of Montana (Running et al., 2004) and rescaled to 0.5°. We also obtained GPP data from VPM (Zhang et al., 2017b) and PML-v2 (Zhang et al., 2019). Comparisons of these GPP datasets are given in Fig. S4-5, which shows that spatially the GPP values from MODIS and VPM are more comparable than PML which provides higher values. The annual trends inferred by the three products vary across the basin, mostly within the range of -25 to 25 $gCm^2 yr^{-1}$. Correlation coefficients between each two GPP datasets are high at both the monthly and annual scales, especially over the areas where croplands predominate However, without extensive gridded ground observations in the basin to validate these datasets, it is hard to conclude which one is most accurate. With the assumption that the ensemble mean values from multiple datasets can effectively reduce data uncertainty lying in an individual dataset, we used the mean GPP from the three sources for further analysis.

Information of data sources, resolution and time span for all variables related to this study is listed in Table 1. To compare with GRACE data, anomalies of P, AI, NDVI, and GPP data were calculated by subtracting the means over the same baseline period of GRACE data (i.e. 01/2004–12/2009). All variables were obtained from 04/2002 to 03/2015 covering 13 hydrological years. Cubic spline interpolation was applied to fill the missing monthly data for the GRACE, MOD16/17 and PML.

**2.3 Data analysis**

To investigate the changes in hydroclimate and vegetation, we carried out trend analysis using the Mann-Kendall (MK) test method both in space and in time. The MK test does not require normality of time series and is less sensitive to outliers and missing values (Pal and Al-Tabbaa, 2009). This non-parametric test method has been used in many studies to detect changing hydrological regimes (Déry and Wood, 2005; Zhang et al., 2009). Interplay between hydroclimate and vegetation was quantified by linear regression; the Pearson correlation coefficient ($r$) and coefficient of determination ($R^2$) were taken

as a measure for assessment of the linkages between different variables. Data series were detrended by removing the linear trends before analysing their relationships at both the monthly and annual scales. Furthermore, to investigate the causal role of vegetation growth to water availability changes (or vice versa), we carried out lag effect analysis between vegetation parameters and hydroclimate variables a lag effect analysis was carried out to determine the temporal dependency between variables where the linear relationship was not obvious. That is, we assume that vegetation growth is constrained by water resources if dynamics of NDVI/GPP falls behind dynamics of P/TWSA.

[revised manuscript text omitted]

betweenin dry and wet years, with 14.39.8% and 6.9% lower and 14.9% higher in dry years for minimum and maximum values, respectively. This implies firstly that vegetation greenness is more sensitive to any changes in hydroclimate than productivity, and secondly that. Moreover, GPP in growing seasons (i.e. October to April in general definition) in dry years was relatively higher than that in wet years reflecting a positive effect of water stress on biomass accumulation.

290 Fig. 10 gives the $R^2$ from linear regression between the monthly climatological means of different variables considering phase shift for lag analysis over all the years, dry and wet years, respectively. It shows NDVI varied strongest with P, TWSA and AI in the previous 3, 1 and 3 months, respectively when considering all data during 2002-2014. In comparison, a shorter lag time of GPP to P, TWSA, and AI was detected (21, 0, 1 month, respectively). Comparison of the lag time in dry and wet years shows that the influence of P on vegetation was more prominent in wet years than in dry years, while TWS influence

295 was greater in dry years than wet years. Moreover, NDVI responded faster to dryness change in dry years (2 months) than wet years (3 months), and GPP responded slower to dryness change in dry years (1 month) than wet years (0 month). This may indicate that drying to some degree can stimulate biomass production. In addition, GPP varied synchronously with TWS showing a high dependency on water storage despite the dryness conditions.

**4 Discussion**

300 **4.1 Uncertainties in the datasets and results**

Data availability is one of the greatest obstacles for large-scale and long-term ecohydrological studies. Remote sensing products are thus useful to characterize ecohydrological changes in a large sparsely monitored basin. In this study, we used remote sensing and assimilated data of water storage, vegetation status and precipitation to assess their relationships. Precipitation is one of the commonly monitored meteorological variables,

305 usually with relatively long time series and wide spatial coverage. We compared P data from GLDAS and meteorological stations in Fig. S1. It shows that the two datasets agree well both spatially and temporally. The spatial coefficients of determination ($R^2$) range from 0.7 to 0.9 in pixels where stations are available, and the temporal $R^2$ is 0.98 with a close-to-one regression slope. The comparison indicates that the gridded GLDAS precipitation data can be used to analyse the dynamics and relationships of hydroclimate and vegetation parameters. Potential evapotranspiration (ETp) is used to

310 calculated aridity index, therefore, we also obtained and compared ETp in Fig. S2-3, which shows that spatially the correlation coefficient between monthly and annual ETp lies mostly in 0.6~1.0 and 0.4~1.0, showing relatively good agreement; and temporally ETp are close to each other at the monthly scale while the uncertainty enlarges at the annual scale. In lack of ground truth data, and with the assumption that ensemble means can reduce the errors in each individual product, we calculated the average ETp from the two sources for analysis.

315 GPP data from MODIS have been extensively used in literature to facilitate studies of vegetation in response to climate and hydrology. For example, A et al. (2017) discussed the relationship between TWS, soil moisture and GPP in response to

[revised manuscript text omitted]

395 2018). Whilst at the monthly scale NDVI was still strongly influenced by TWS but not so strongly by P, in comparison to the strong response of monthly GPP to both P and TWS. The weakened linear influence of P on NDVI at the monthly scale, found also by others such as Bai et al. (2019) and A et al. (2017), can be explained by the lag effect that NDVI lagged by 3 and 1 months after P and TWS, respectively. In comparison, the lag time between GPP, P and TWS was 2 and 1 month shorter than NDVI versus P and TWS (Fig. 10a). The differences in NDVI and GPP response to hydroclimate variables may

400 lie in the way these two parameters are calculated, especially that GPP is calculated based on atmospheric variables like temperature, vapor pressure deficit and photosynthetically active radiation (Pei et al., 2020). Because of the asynchrony in the atmospheric variables and NDVI (Piao et al., 2006), the GPP and NDVI would also have some inconsistency in time. This would further indicate that it should be given more caution when choosing parameter (NDVI or GPP) to better representflect vegetation growing status, which is lack in literature for discussion.

405 In addition, comparison of the plant-water relations in dry and wet years showed a slower response of GPP to aridity index in dry years than wet years (Fig. 10b-c). Wilcoxon rank sum test shows that the areal mean NDVI and GPP in dry years are not significantly different from those in wet years ($p$=0.12 and 0.76) (Fig. 9c-d). In fact, GPP was higher in the growing seasons in dry years than wet years, and NDVI was lower in non-growing seasons of dry years than wet years. Together, these comparisons may imply that a certain degree of drying can stimulate biomass accumulation. This phenomenon is also

410 revealed by other studies (Zhang and Zhang, 2019). The underlying mechanisms could be similar to the principle of regulated irrigation in agricultureal practice to increase water use efficiency under a certain degree of water stress (Chai et al., 2016), or that the atmospheric conditions are more favourable for photosynthesis during dry years than wet years (Restrepo-Coupe et al., 2013; Zhang and Zhang, 2019), given that the soil water or groundwater storage is not depleted severely in these dry years. This dryness effect on ecosystem productivity cannot be detected in the annual scale assessment (Brookshire

415 and Weaver, 2015; Yao et al., 2020). These results indicate firstly that pre-growing season hydroclimate conditions play a

key role in the follow-on vegetation growth and production (Wang et al., 2019), and secondly that water limits vegetation even in this subtropical radiation- and rain-abundant region instead of water shortage resulted from vegetation establishment. It cannot be detected at the time scales investigated in this study that vegetation consumes excessive water through transpiration that results in obvious reduction in water storage. However, 
[revised manuscript text omitted]

[Figure]

**Figure 8**. Scatter plot of monthly anomalies of precipitation (P), total water storage (TWS), aridity index (AI), NDVI and GPP.

[Figure]

**Figure 9**. (a-b) Monthly variations of anomalies of precipitation (P), total water storage (TWS), aridity index (AI, scaled for
770   a better view), and NDVI, gross primary production (GPP) in all years; (c) monthly means of dry hydrological years and (d)
monthly means of wet hydrological years during 2002-2014. Plots *c* and *d* share the same units and legends with plots *a* and
*b*. Shaded areas show the standard errors of each variable.

[Figure]

775 **Figure 10.** Coefficient of determination between monthly anomalies of precipitation (P), total water storage (TWS), aridity index (AI) and NDVI and GPP in (a) all years, (b) the dry years, and (c) the wet years after shifting different number of months.

---

## Author Response (AR2)

We thank the reviewer again for the detailed and constructive comments that help improve our manuscript greatly in several aspects.

**RC1:** *The quality of the revision is not satisfactory. I recommend major revision rather than rejection mainly because of the relatively strong yearly correlation between NDVI and GRACE TWS ($R^2$~0.59), which if carefully analyzed has the potential to lead to informative conclusion regarding regional ecohydrology in a humid area. Claiming water limitation in such a humid area needs caution. The Pearl River basin features strong rainfall that is generally higher than the potential evapotranspiration (i.e., rainfall outweighs water demand) and the river runoff is large, too. The claim about water limitation therefore needs justification and explanation. When and where exactly does water limitation come from, is there a period in each year when the water supply cannot meet the demand of plant growth? The authors have multiple datasets (both water supply and demand) available to answer these questions.*

**AC1:** We have adopted the advice of using EVI and SIF instead of NDVI for the analysis, and therefore, rewrote some parts of the manuscript in each section. Discussion has also been rearranged with the uncertainty moved to Methods section. With some thoughts and trials, we decided not to adopt the suggestion of using the entire study period as the baseline to calculate the anomalies though, and reasons are given in the response to that specific comment. We want the reviewer to know that we take the comments and suggestions with gratefulness, and carefully addressed them to the best of our knowledge.

The critical questions mentioned above are important in this study and have been carefully addressed where possible with support from the data and stated in the relevant sections in Results and Discussion. The annual water availability is relatively high in this monsoonal humid subtropical basin, but this does not guarantee no water stress to plants due to the obvious seasonality of both water availability and vegetation growth. Our best guess is that the water stress could be a result of long growing seasons since majority of the forests are evergreen and the crops are planted on rotations throughout the year. Irrigation water needs to be added in croplands to supplement water supply in addition to rainfall during dry seasons. In addition, radiation energy is intermittently available to plants because of the cloudy/rainy conditions during the growing seasons and its periodic increases will improve vegetation productivity. Overall, using EVI, SIF and GPP for analysis instead of NDVI, we found the water supply did not influence greenness as much as productivity, supported by the long-term mean monthly data in Fig. 8c-d. Please refer to the revised m/s.

**RC2:** *I suggest the authors use EVI and SIF both of which are easily available. The authors' argument is problematic as most of the significant trends in NDVI occur in the central portion of the study domain where notable forest cover exists (Fig. 1c & 4b). Using EVI can avoid the saturation issue associated with NDVI. In addition, the adopted GPP products may not represent well the soil moisture constraint. Using SIF will provide an observational metric that represents GPP.*

**AC2**: While EVI can avoid the light saturation problems, taking the advice, we use EVI, GPP and SIF datasets in the analysis in this version. MODIS EVI shows difference in dynamics compared to GIMMS NDVI, showing better concurrency with GPP and SIF overall. Because of the GOSIF algorithm and input data (OCO-2, MODIS, etc.), it has high correlation with MODIS EVI ($R^2$=0.95), which leads to similar relationships between hydroclimate and EVI and SIF.

Using the new datasets in the analysis and taking out the analysis with NDVI, we have updated the results and discussion accordingly. We think the different vegetation indices are worth a sophisticated comparison regarding phenological indication and ecohydrological applications, and we found some papers have done some work on it. We will pursue farther studies in separate works on this matter.

**RC3:** *Line 20. The authors conclude that "the degree of water restriction on vegetation was higher than that of water consumption by vegetation even in this rain-abundant region." It is still unclear to me how water restriction (which is quantified here by correlation) can be directly compared with plant water consumption (which is often quantified by transpiration).*

**AC3:** This statement was a bit confusing and may not be expressed properly by 'water restriction' and 'water consumption'. We have rewritten it.

This conclusion originates in the assumption that if vegetation parameters' series change ahead of water supply (e.g., TWS) then the negative impacts of vegetation on water supply outweigh the positive impacts of water on vegetation growth, and vice versa. From the analysis we observed the leading role of water supply variations rather than vegetation parameters, based on which we concluded that water supply leads vegetation development, and that vegetation is not dominant in reducing water availability. Therefore, it is more of a qualitive conclusion. Using plant transpiration to compare with either P or TWS can show the proportion of water consumption by vegetation, and the temporal variations can demonstrate which one leads the changes. As in Kirchner et al., (2020) and Deutscher et al., (2016) who showed the diurnal variations of sap flow, groundwater and streamflow and attributed hydrologic variations to plant water use. The idea in this study follows their conclusion in that way. We hope the discussion part makes sense to you.

Kirchner, J., Godsey, S., Osterhuber, R., McConnell, J. and Penna, D.: The pulse of a montane ecosystem: coupled daily cycles in solar flux, snowmelt, transpiration, groundwater, and streamflow at Sagehen and Independence Creeks, Sierra Nevada, USA, Hydrol. Earth Syst. Sci. Discuss., 1–46

Deutscher, J., Kupec, P., Dundek, P., Holík, L., Machala, M. and Urban, J.: Diurnal dynamics of streamflow in an upland forested micro-watershed during short precipitation-free periods is altered by tree sap flow, Hydrol. Process., 30(13), 2042–2049

**RC4:** *Line 118-119. The description about GRACE uncertainty is not accurate. The leakage error depends on the specific basin of interest and needs to be calculated case by case, and I don't think they are released by the processing centers. Wiese et al 2016 documented the recommended procedure to estimate the leakage error (although the values provided in that paper were for RL05). How did the authors quantify the leakage error for the Pearl River basin? Also note that the mascon uncertainties provided by JPL are uncertainties associated with each mascon estimate (not the uncertainty associated with a single 0.5 degree pixel). Based on the short description provided by the authors, it is difficult to assess whether the GRACE uncertainty is treated properly.*

**AC4:** We totally agree it is important to estimate data uncertainties for any quantitative studies. The JPL provides both the measurement errors and leakage errors for the 1-degree GRACE data. Save et al., (2016) stated that quantifying leakage errors does not impact CSR mascon solutions as much as it affects JPL mascon estimate due to the native estimation resolution of $1°$ for CSR mascons versus $3°$ for JPL mascons. Anyway, unluckily no such error attributes are provided in the 0.5-degree GRACE data based on the mascons solutions.

We do not intend to estimate leakage errors following Wiese et al., 2016 method, because we primarily used TWSA to infer water availability changes, so it is the dynamics and trends, not the absolute values, that really matter. In the case of water balance studies or other quantitative hydrologic studies such as using it to verify water storage simulation results, it is very necessary to estimate the data errors. That is not the case of this study.

Here, we simply used the standard deviation of the time series to represent the uncertainty of the data, i.e. the standard deviation of TWSA in each year represents the spatial variability of TWSA in that year across the basin. We have made our point regarding the data uncertainty more clearly in the Methods section to avoid further confusion and argument.

Save, Bettadpur, and Tapley, 2016. High-resolution CSR GRACE RL05 mascons, JGR-Solid Earth, 121 (10): 7547-7569

**RC5:** *Line 175-176 and Fig 2a-c. As I pointed out in my previous comments, the TWSA depends on the choice of the reference period. The common practice is to use the entire analyzed period to represent the "normal" condition. I suggest the authors do the same because using the 2004-2009 as the reference period is not the best effort practice when longer baseline exists, unless the authors have specific reasons to believe the 2004-2009 period better represents the normal condition. This would potentially affect Fig 2a-c in the manuscript and also the classification of dry and wet years in Figs 9 and 10. The authors' response to my original comment stressed that the reference period wouldn't affect the trend analysis, which was obvious and not relevant to my comment.*

**AC5:** There are two ways of processing other data series to comply with GRACE data. One is to use 2004-2009 as the baseline period suggested by JPL Tellus, and the other is to use the entire study period as suggested. We adopted the first one.

We tried with the entire period (2002.04-2015.03). We found firstly that the dynamics (highs and lows) and the linear trend of the newly calculated 'anomaly' are the same with that using 2004-2009; and secondly, the mean value of the new annual 'anomaly' for each pixel is nearly zero when we used the already anomaly data provided by JPL and CSR to subtract their means. This alters the original concept of data anomaly, and it is more of the anomaly of anomaly. Similar studies commonly use the level-3 anomaly products. If we had the absolute mass field, then we can calculate anomaly regarding any baseline period, because it then would be essentially just a matter of different subtractors. But the time-mean fields that are removed in the processing represent essentially the Earth's mean gravity field. As such, there isn't really a 'time-mean mass field', particularly not in terms of average water storage height (https://grace.jpl.nasa.gov/about/faq/).

Using a different baseline does not affect the identification of dry and wet years because we considered the annual variations of TWSA, vegetation index as well aridity index when doing this. It will change the TWSA magnitude little since the period 04/2002-03/2015 is not significantly longer than 01/2004-12/2009. However, the pixelwise mean annual newly calculated anomaly is close to zero, which is not helpful in detecting the hotspot of water availability changes.

For the reasons above, we kept the 2004-2009 baseline period to calculate the anomalies of other data series.

**RC6:** *Figure 7. It is not appropriate to calculate correlation with TWS at pixel level as the TWS resolution is intrinsically coarse (~ 3 degree) and going beyond the mascon resolution will rely on scaling factors that are not based on pure observations. This should be either noted or avoided. The basin scale results shown in Fig 6 are more appropriate.*

**AC6:** Thanks for pointing this out. We agree. A few sentences regarding this matter have been added in the first paragraph of section 3.1 and 3.3. We keep it mainly because it can show that the significant relationships mostly exist in the central cropland areas.

**RC7:** *Line 226. As I asked in my previous comment, please clarify if the seasonality has been removed when calculating correlation using the monthly time series. Note that to quantify water limitation, the seasonality should be removed from the monthly time series.*

**AC7:** Description has been made clearer in the Data and Methods section 2.4. We have firstly removed the linear trend from the data, and then removed the seasonality by subtracting the seasonal signals (i.e. the monthly climatological means) from the detrended data.

**RC8:** *Figure 8. Is each point corresponding to a pixel? Note that this is not appropriate for TWS. See my comment as above.*

**AC8:** Each point represents a monthly data averaged over the entire basin, not each pixel. We have made it clear in the caption and the text related with the figure.

**RC9:** *Line 188-190. It is difficult to infer causality between two variables simply based on temporal lag. The drop in NDVI occurs after Nov. Is this period also part of the growing season? Is there any energy limitation? When the NDVI starts to decline, does water demand outweigh water supply?*

**AC9:** Please refer to the response to comment RC3 and RC2 above. Monthly climatological mean EVI shows different characteristics from NDVI in a way that it synchronizes better with other vegetation parameters and water components. Some of the phenomena mentioned is gone.

**RC10:** Figure 9c-d. The range of the right y-axis is way too large for AI and NDVI. -0.5 to 0.5 is likely more appropriate.

**AC10:** Thanks for the advice. We have modified the y-axis range to make the variations more visible. Now it is within the range of -0.4 to 0.5.

**RC11:** *Line 240. If the difference in NDVI between the dry and wet years is mainly attributed to the difference in the non-growing season, does that mean the growing season NDVI does not show significant difference between dry and wet years? That would indicate that water is not a limiting factor to growth in the study domain.*

**AC11:** With EVI in the analysis, the difference between dry and wet years is not as obvious as indicated by NDVI, although the EVI anomaly is still slightly lower in dry years than in wet years in both growing and non-growing seasons.

Previously, analysis with NDVI indicates that the difference lies primarily in non-growing seasons between dry and wet years, while the difference is not so obvious in growing seasons. This does not infer that vegetation is not limited by water, and the main reason could be that in dry years irrigation water is often supplemented to secure food production in the croplands as we argued. The current analysis did not account for the additional water supply but only the natural precipitation and water storage.

In this revision, we have carefully rewritten the relevant part in the results and discussion to make sure them consistent.

**RC12:** *Line 246-247. That seems contradictory to the notion of water limitation.*

**AC12:** This can be partly referred to the response to the comment RC3. After a second thought, we feel the expression should be improved in the revision. Our intention was to state that under dry conditions (with less precipitation/storage), water surplus by irrigation plays an important role in boosting plant production.

**RC13:** *Line 248 vs. Fig 10. Is it "monthly climatological mean" (Line 248) or is it "monthly anomalies" (Fig 10 caption)?*

**AC13:** After doublechecking the code, we confirm that it should be monthly climatological mean of the data anomaly. The caption has been corrected. Thanks!

**RC14:** *Section 4.1 As I pointed out in the original comment, this section should go to the Method (some of them should go to introduction, e.g., Line 259-261, 272-280). The discussion should focus on the scientific hypothesis that the authors aim to address. I believe the other reviewer pointed out the same issue as I did.*

**AC14:** We have now moved this part to Methods and Introduction where appropriate. By doing that, we edited the Data and Methods section to avoid repetition and for a better logic of flow.

**RC15:** *Line 314. The reason that I pointed out Tong et al., 2018 in the original comment is to remind the authors that there is an active human intervention in the study (with respect to land use and land cover change), which would very likely complicate the explanation of plant-water relation – in this case, the increase in growth is partly attributed to human management rather than plant response to water supply changes.*

**AC15:** Yes, in this version of discussion, we also directed our findings to human intervention in section 4.1 corresponding to your kind reminder. Presumably, these interventions are mainly the irrigation activities and possible planting structure adjustment during dry periods. Thus, the plant-water relationships might not be under purely natural conditions but also exposed to anthropogenic influences.

---

## Author Response (AR3)

*Editor's comment to the Author:*

*We have now received the reports of the two reviewers with comments on the revised manuscript. Though one of the reviewers is very positive, the other still has some concerns. After my own assessment of the manuscript and the reviewers' comments, I agree with Reviewer #1 and believe that the manuscript needs further revisions to clarify some methodological aspects, and their implications for the analysis and results.*

*It is important to evaluate/discuss the possible implications of leakage errors (section 2.2). Please also include further analysis for the selection of the 2004-2009 period and its representation of normal water conditions over the domain. Addressing these comments will further improve the value of the manuscript contributions.*

Thank you for the assessment of our manuscript and provision of the comments. The manuscript cannot be improved so much without the help from the reviewers and yourself.

We have revised the manuscript to describe how 'normal' the period 2004-2009 compared to the entire period using the annual precipitation data in Line 147-153: mean annual P was 1444.0±138.0 mm over the period of 2004-2009, comparable to 1461.3±150.7 mm over the entire study period, which means that the period of 2004-2009 is representative of the normal condition over the study period.

Regarding the leakage errors along the basin boundary, we have added more information in the section 2.2 about GRACE data in Line 123-133.

We hope that with the newly added clarification, the manuscript can meet the standard of the journal in terms of the rigor in describing the data, methods and results. Sincere thanks to you all!

*Reviewer #1*

*RC1: The authors argue that they do not intend to estimate leakage errors following Wiese et al., 2016, because they primarily used TWSA to infer water availability changes instead of the absolute values.*

*This logic is flawed. Method from Wiese et al 2016 has been widely used to quantify leakage error related to TWS variability and trend. To my knowledge most studies adopt their method to address leakage error associated with trends. For this study, the region to the north of the study domain is known to have experienced sustained increase in TWS linked to reservoir filling, which makes it necessary to evaluate if any of those signals have leaked into the analyzed domain.*

AC1: First of all, we want to confirm that we agree that the measurement and leakage errors are important when using GRACE data for quantitative analysis. Many studies have discussed the methods for error estimation such as Long et al., (2017) and Wiese et al., (2016), although in many hydrologic applications the error estimations are not described such as in Muskett et al., (2009) and Yang et al., (2014).

We read carefully again the Wiese et al., 2016 paper and double checked the product

information of GRACE data (Release 6). Wiese et al., 2016 paper states clearly that their procedures to reduce leakage errors across the land/ocean boundary are "unique to the JPL RL05M mascon solution and are not directly applicable to other GRACE mascon solutions. The reason for this is that JPL RL05M is currently the only available mascon solution that parameterizes the gravity field in terms of equal-area 3° spherical cap mascons. Other available mascon solutions parameterize the gravity field in terms of a finite spherical harmonic expansion of 1° mascon elements". The CSR RL06 mascon solution uses a grid of 1-degree and the hexagonal tiles that span across the coastline are split into two tiles along the coastline to minimize the leakage between land and ocean signals (Product Highlights at http://www2.csr.utexas.edu/grace/RL06_mascons.html). The CSR RL06 data include all necessary corrections (http://www2.csr.utexas.edu/grace/RL06_mascons.html). The JPL RL06 uses mascon grids of 3-degree, and scaling factors derived from CLM land surface model at 0.5-degree resolution are provided which are multiplied to calculate the TWSA. We then checked the JPL RL06 product information in the netcdf file, and found that apparently this RL06 product has already incorporated the coastline leakage error provided by Wiese et al. ,2016 method (see below).

```
months_missing              = '2002-06;2002-07;2003-06;2011-01;2011-06;2012-05;2012-10;2013-03;2013-08;2013-09;2014-02;2014-07;2014-12;2015-06;2015-10;2015-11;2016-04;2016-09;2016-10;2017-02'
postprocess_1               = ' OCEAN_ATMOSPHERE_DEALIAS_MODEL (GAD), MONTHLY_AVE, ADDED BACK TO OCEAN PIXELS ONLY'
postprocess_2               = 'Water density used to convert to equivalent water height: 1000 kg/m^3'
postprocess_3               = 'Coastline Resolution Improvement (CRI) filter has been applied to separate land/ocean mass within mascons that span coastlines'
GIA_removed                 = 'ICE6G-D; Peltier, W. R., D. F. Argus, and R. Drummond (2018) Comment on the paper by Purcell et al. 2016 entitled An assessment of ICE-6G_C (VM5a) glacial isostatic adjustment model
geocenter_correction        = 'Swenson, Chambers, and Wahr (2008), J. Geophys. Res., 113, 8410.'
C_20_substitution           = 'Cheng, M., Ries, and Tapley (2011), J. Geophys. Res., 116, B01409.'
user_note_1                 = 'The accelerometer on the GRACE-B spacecraft was turned off after August 2016. After this date, the accelerometer on GRACE-A was used to derive the non-gravitational accelerations a
journal_reference           = 'Watkins, M. M., D. N. Wiese, D.-N. Yuan, C. Boening, and F. W. Landerer (2015) Improved methods for observing Earth's time variable mass distribution with GRACE using spherical cap
CRI_filter_journal_reference = 'Wiese, D. N., F. W. Landerer, and M. M. Watkins (2016) Quantifying and reducing leakage errors in the JPL RL05M GRACE mascon solution, Water Resour. Res., 52, doi:10.1002/2016WR0193
```

In this case, we have added a few more sentences to include the above information for a better description of how errors in the GRACE TWSA are processed. They are given in line 123-133.

Long, D., Pan, Y., Zhou, J., Chen, Y., Hou, X., Hong, Y., et al. (2017). Global analysis of spatiotemporal variability in merged total water storage changes using multiple GRACE products and global hydrological models. *Remote Sensing of Environment*, *192*, 198–216. https://doi.org/10.1016/j.rse.2017.02.011

Muskett, R. R., & Romanovsky, V. E. (2009). Groundwater storage changes in arctic permafrost watersheds from GRACE and insitu measurements. *Environmental Research Letters*, *4*(4). https://doi.org/10.1088/1748-9326/4/4/045009

Yang, Y., Long, D., Guan, H., Scanlon, B. R., Simmons, C. T., Jiang, L., & Xu, X. (2014). GRACE satellite observed hydrological controls on interannual and seasonal variability in surface greenness over mainland Australia. *Journal of Geophysical Research: Biogeosciences*, *119*(12), 2245–2260. https://doi.org/10.1002/2014JG002670

*RC2: The authors justify their use of 2004-2009 baseline using the following argument: (a) the 2004-2009 baseline is "suggested by JPL," (b) calculating anomalies using the entire period as the new baseline is "more of the anomaly of anomaly," (c) the entire period is not significantly longer than the 2004-2009 period, and (d) the newly calculated mean annual anomaly is close to zero.*

*I want to first stress that for the purpose of identifying anomalous dry or wet conditions,*

*the suitability of a certain choice of baseline should be evaluated based on how normal the water conditions are during that baseline period. In practice longer baseline is often chosen to average out anomalies across different time scales.*

*The authors did not evaluate how normal the water conditions were during the 2004-2009 period but instead provided four problematic arguments. (a) The 2004-2009 baseline is indeed used by JPL since earlier releases of the GRACE data but claiming that the 2004-2009 baseline is suggested by JPL would be of a different nature. In fact, JPL has provided guidelines regarding how to adopt a different baseline on the same link provided by the authors in the rebuttal letter. (b) Removing the mean over the entire period do not provide results that correspond to an anomaly of anomaly; they are simply anomalies referenced to a new baseline period. (c) The duration of the entire period is more than twice as that of the 2004-2009 period. (d) This sounds peculiar. Shifting the baseline will increase the magnitude of some of the anomalies while reducing the rest.*

*Finally, if the authors insist using the shorter period as the baseline, they should clarify how well that shorter period represents the normal water condition in the analyze domain.*

AC2: We would like to thank you for such detailed comments again. With all respect, we keep the 2004-2009 as the baseline for all variables for the reason below.

The anomaly is the value deviating from a reference. If we had the true absolute water storage value for each month, then we can use any period as baseline reference to calculate the anomaly; but GRACE products are already storage anomaly (without absolute storage values). Therefore, subtracting the GRACE monthly anomaly data by the all-time mean, which is what we meant 'the anomaly of anomaly', will lead to near-zero results.

Anyway, the calculations with GRACE data show that the monthly and annual data using 2004-2009 baseline and entire period baseline have slight difference with the former having generally larger values than the latter (Fig. 1 a-b). So, from this view, it has no problem with the entire period as baseline.

[Figure]

Fig. 1 (a-b) Comparison of TWSA calculated using two baseline periods at the monthly and annual scale. Data points include all GRACE data in the study area over 156 months and 13 years respectively. (c-d) Mean water storage anomaly over the entire study area with two baseline periods. Dashed lines are the all-time average over the area.

However, when we looked at the all-time mean TWSA over the entire study area, we found the mean value with baseline of entire period is nearly zero (Fig. 1 c-d). This is also true for the average value for each grid with baseline of the entire period (Fig. 2), that is, the all-year mean value for each grid is nearly zero, although the annual trends basically are the same with these calculated with baseline of 2004-2009.

[Figure]

Fig. 2 Spatial distribution of men annual TWSA calculated with baseline of the entire study, and the trends.

Putting that aside, we calculate the annual mean precipitation of the basin and found it was 1461.3±150.7 mm over the entire period, comparable to 1444.0±138.0 mm over 2004-2009. The means that the period of 2004-2009 can be representative of the normal condition. We have added this explanation in the revision (line 147-153).

[Figure]

Fig. 3 Variations of mean annual precipitation over the basin. Red and blue lines in the figure mark the average value over the entire period and over 2004-2009, respectively.